# Unraveling the mechanism of small molecule induced activation of *Staphylococcus aureus* signal peptidase IB
Shu-Yu Chen [1,2,5], Michaela K. Fiedler [3,5], Thomas F. Gronauer [3], Olesia Omelko [3], Marie-Kristin von Wrisberg[4], Tao Wang[3], Sabine Schneider [4], Stephan A. Sieber [3] ✉ & Martin Zacharias [2] ✉

*Staphylococcus aureus* signal peptidase IB (SpsB) is an essential enzyme for protein secretion. While inhibition of its activity by small molecules is a well-precedented mechanism to kill bacteria, the mode of activation is however less understood. We here investigate the activation mechanism of a recently introduced activator, the antibiotic compound PK150, and demonstrate by combined experimental and Molecular Dynamics (MD) simulation studies a unique principle of enzyme stimulation. Mass spectrometric studies with an affinity-based probe of PK150 unravel the binding site of PK150 in SpsB which is used as a starting point for MD simulations. Our model shows the localization of the molecule in an allosteric pocket next to the active site which shields the catalytic dyad from excess water that destabilizes the catalytic geometry. This mechanism is validated by the placement of mutations aligning the binding pocket of PK150. While the mutants retain turnover of the SpsB substrate, no stimulation of activity is observed upon PK150 addition. Overall, our study elucidates a previously little investigated mechanism of enzyme activation and serves as a starting point for the development of future enzyme activators.

Classical drug development primarily focuses on compounds that shut down pathogenesis-associated pathways e.g. by inhibition of enzyme activities[1]. Challenges for such an inhibitor design often comprise the need for high target occupancy and at the same time limit the drug exposure to avoid negative side effects[1–4]. In contrast, enzyme activation exhibits several advantages over inhibition with increases of activity by only 10–20% sufficient to show phenotypic effects, alleviating the need for high occupancy binders and keeping the exposure low[1,5]. Despite these intriguing perspectives, only two activators gained overall FDA approval over the course of the past five years, highlighting an unexplored therapeutic potential compared to the prevalent focus on inhibitors[6–11].

One reason for this shortage is the challenge in activator design which is not only based on mere binding but also on conformational changes to stimulate turnover. Thus, most activators are discovered by coincidence or activity screens and often lack a firm mechanistic understanding of their mode of activation. However, these in-depth mechanistic analyses are pivotal to advance the rational design of activators and catalyze their use in drug development[1,12–15].

One prominent example of treating bacterial infections via enzyme activation are acyldepsipeptides (ADEPs) which exert their unique anti-bacterial activity by targeting the caseinolytic protease P (ClpP). ADEPs bind allosterically to ClpP opening its axial pore and converting it from a

[1]Department of Chemistry and Applied Biosciences, ETH Zurich, Vladimir-Prelog-Weg 2, Zurich, 8093, Switzerland. [2]TUM School of Natural Sciences, Department Biosciences, Theoretical Biophysics (T38), Center for Functional Protein Assemblies (CPA), Technical University Munich (TUM), Ernst-Otto-Fischer Str. 8, Garching, 85748, Germany. [3]TUM School of Natural Sciences, Department Biosciences, Chair of Organic Chemistry II, Center for Functional Protein Assemblies (CPA), Technical University Munich (TUM), Ernst-Otto-Fischer Str. 8, Garching, 85748, Germany. [4]Department of Chemistry, Ludwig-Maximilians University Munich (LMU), Butenandtstr. 5-13, Munich, 81377, Germany. [5]These authors contributed equally: Shu-Yu Chen, Michaela K. Fiedler. ✉e-mail: stephan.sieber@tum.de; zacharias@tum.de

tightly regulated peptidase to an uncontrolled chaperone-independent proteolytic machinery[16–18].

This mechanism has deleterious effects on the cell, as essential proteins required for cell integrity are digested by the activated protease. In addition, ADEPs effectively kill difficult-to-treat persister cells[19], likely because stimulation of ClpP turns on uncontrolled proteolysis and consequently self-digestion of dormant bacterial cells.

We recently discovered another example of antibiotic-mediated peptidase stimulation when investigating the mode of action (MoA) of the potent antibiotic PK150[20]. Target identification by chemical proteomics in *Staphylococcus aureus* revealed signal peptidase IB (SpsB) and menaquinone biosynthesis methyltransferase (MenG) as major hits. SpsB (Uniprot ID: Q2FZT7) is an essential membrane-bound serine-endopeptidase involved in the cleavage of the amino-terminal signal peptides of secreted bacterial pre-proteins utilizing the unique Ser/Lys catalytic dyad[21–23]. Mechanistically, serine (S36) acts as the acylating nucleophile, and lysine (K77) is the general base in both the acylation and deacylation step of catalysis. In the first step, K77 forms a catalytic H-bond with the acylating serine nucleophile (S36). Upon deprotonation, S36 attacks the amide group of the substrate forming tetrahedral intermediate I. The leaving group amide is protonated by K77 releasing the mature protein and the acyl-enzyme intermediate is formed. For hydrolysis, water enters the active site and forms a second tetrahedral intermediate to release the cleaved signal peptide, regenerating the SpsB active site[22,24–27].

Target validation showed that PK150 induces an up to 3-fold over-activation of SpsB having a phenotypic effect on *S. aureus* leading to uncontrolled secretion of proteins including members of the autolysin family. Dysregulation of autolysin abundance is known to induce cell-wall degradation which was confirmed by mass spectrometry (MS), electron microscopy (EM), and autolysis experiments[20]. Therefore, stimulation of SpsB activity represents a promising strategy for drug development against MRSA. Putative crucial residues for activation were hypothesized, however, its precise mechanism of activation remained elusive[20].

Here, we identified the binding site of PK150 in SpsB experimentally via mass spectrometry and elucidated the mechanism of activation by subsequent Molecular Dynamics (MD) simulations. The compound binds into an allosteric pocket near the active site and thereby shields the catalytic dyad from excess water exposure. Excess water causes increased hydration of the catalytic residues and reduced formation of the hydrogen bond between S36 and K77 necessary for catalysis. The proposed binding mode and mechanism was confirmed by mutating key residues aligning the allosteric pocket. While the mutants retained SpsB activity, compound-induced stimulation of turnover was significantly impaired. Overall, this study illustrates a novel principle of overactivation which serves as inspiration and rationale for a future generation of enzyme activators.

## Results
### Molecular dynamics simulations reveal a substrate and an allosteric pocket near the catalytic center

*S. aureus* SpsB is a membrane-bound protein consisting of a transmembrane helix (membrane domain) and a large globular domain on the extracellular side (extracellular domain). To study the dynamics of the enzyme in its membrane environment, we modeled the 3D structure of the enzyme and performed MD simulations in the substrate-bound (holo) and substrate-free (apo) forms in the dimyristoyl phosphatidylglycerol (DMPG) bilayer (Fig. 1a). Within this structure model, the substrate (GGGGGAP-TAKAA*SK, * is the scissile bond) formed a hybrid β-sheet with the amino acid residues T31-K33 and D74-V76 while the scissile bond was exposed to the catalytic residues S36 and K77 (Supplementary Fig. S1). Simulations of both apo and holo SpsB showed a similar flexibility profile and the residues located on β-strands possess the lowest mobility (Supplementary Fig. S2).

In these simulations, water molecules were frequently found around the catalytic dyad S36 and K77 in the active site. In the apo simulations, the water molecules clustered mostly in two peripheral pockets, hereafter referred to as the substrate and allosteric pocket (Fig. 1b, c). The substrate pocket overlaps with the substrate-binding channel and intersects with the

allosteric pocket at the catalytic center (Fig. 1c). To characterize the dynamics of the pocket, the pocket volumes were measured using a grid-based method implemented in MDpocket[28]. Upon substrate association, the volume of the allosteric pocket and the substrate pocket expanded from $229.04 \pm 117.19$ Å$^3$ to $290.75 \pm 101.62$ Å$^3$, and from $256.58 \pm 105.71$ Å$^3$ to $417.42 \pm 86.69$ Å$^3$, respectively (Fig. 1c). To understand the molecular basis of the breathing motion of the allosteric pocket, three aromatic residues F67, Y75, and F158 close to the catalytic dyad, which can form intramolecular π-stacking, were identified (Fig. 1d). Indeed, the volume of the allosteric pocket correlated moderately with the ring-to-ring distance from F158 to F67 or Y75 (Pearson R = ~0.25–0.48, Supplementary Fig. S3). In MD simulations of apo SpsB, the catalytic center remained in a hydrated environment due to the access of water from both pockets leading to less frequent hydrogen bonds (18.9%), characterized by the distance between the S36 hydroxyl group and the unprotonated K77 amine group of less than 2.5 Å (Fig. 1d, e). In contrast, the catalytic center was only accessed by water molecules through the allosteric pocket in simulations of the holo-enzyme and the catalytic hydrogen bond is more frequently formed (84.1%) (Fig. 1b, f). Notably, our data indicates that a larger S36-K77 distance correlates with a higher K77 water accessibility (Fig. 1f), suggesting that decreasing the water accessibility to the active site is a potential way to stabilize the catalytic geometry.

### Binding of PK150 reduces water accessibility to the catalytic center

PK150 is a recently discovered antibiotic that binds to SpsB and enhances the enzyme activity. A strong correlation between biological activity and SpsB activation was previously identified leading to a phenotypic effect in *S. aureus,* eventually leading to cell death[20]. To elucidate the activation mechanism of PK150 (Fig. 2a), a chemical proteomics workflow using isoDTB (isotopically labeled desthiobiotin azide) tags was first performed to directly decipher the binding site via mass-spectrometry (MS) (Fig. 2b)[29,30]. Herein, PK150-P, an antibiotically active photo crosslinker derivative of PK150, previously used for target identification of SpsB in *S. aureus* (Fig. 2a)[20], was incubated with recombinantly expressed maltose-binding-protein (MBP) tagged SpsB (Supplementary Fig. S4) and irradiated with UV light to crosslink the protein to PK150-P. Heavy and light isoDTB tags were clicked to the alkyne handle of PK150-P by copper-catalyzed *Click* Chemistry. After enrichment of modified peptides on streptavidin beads, samples were analyzed by LC-MS/MS and evaluated using the MSFragger-based Fragpipe computational platform[31–35]. This experiment revealed the binding of PK150-P to the peptide sequence GNVVV*FHANK modified at the residue V66 (Supplementary Tables S2, S3). Of note, this binding site substantially deviates from our previous prediction based solely on docking highlighting the value of combined experimental and theoretical studies[20].

Based on these results, we docked PK150, which fits sterically well into the identified allosteric pocket of the holo enzyme close to V66, and studied its dynamics using MD simulations (Fig. 2c). PK150 bears two characteristic structural units: an aromatic ring with a hydrophobic 4-chloro-3-trifluoromethyl substitution pattern (ring 1) and another aromatic ring with an acetal and a hydrophilic difluoromethylene motif (ring 2, Fig. 2c). The docking position was generated by placing the trifluoromethyl group of ring 1 towards the interior of the protein and ring 2 towards the water environment matching our binding site identification results with PK150-P (Fig. 2c). In this docking pose, ring 1, which was shown to play an essential role in the activation effect[20], is sandwiched by F67, Y75, and F158. Moreover, the central urea moiety linking both aromatic rings forms hydrogen bonds with Q165 (Fig. 2c), that might have an additional important stabilizing effect. No significant change in enzyme flexibility was observed when compared to the ligand-free system (Supplementary Fig. S5). MD simulations suggest that placing PK150 in the allosteric pocket reduces the water accessibility to the catalytic center and thus the hydrogen bond distance between S36 and K77 (Fig. 2d). Additionally, the size of the allosteric pocket increases to approximately 400 Å$^3$ to accommodate PK150 without altering the volume of the substrate pocket (Fig. 2d).

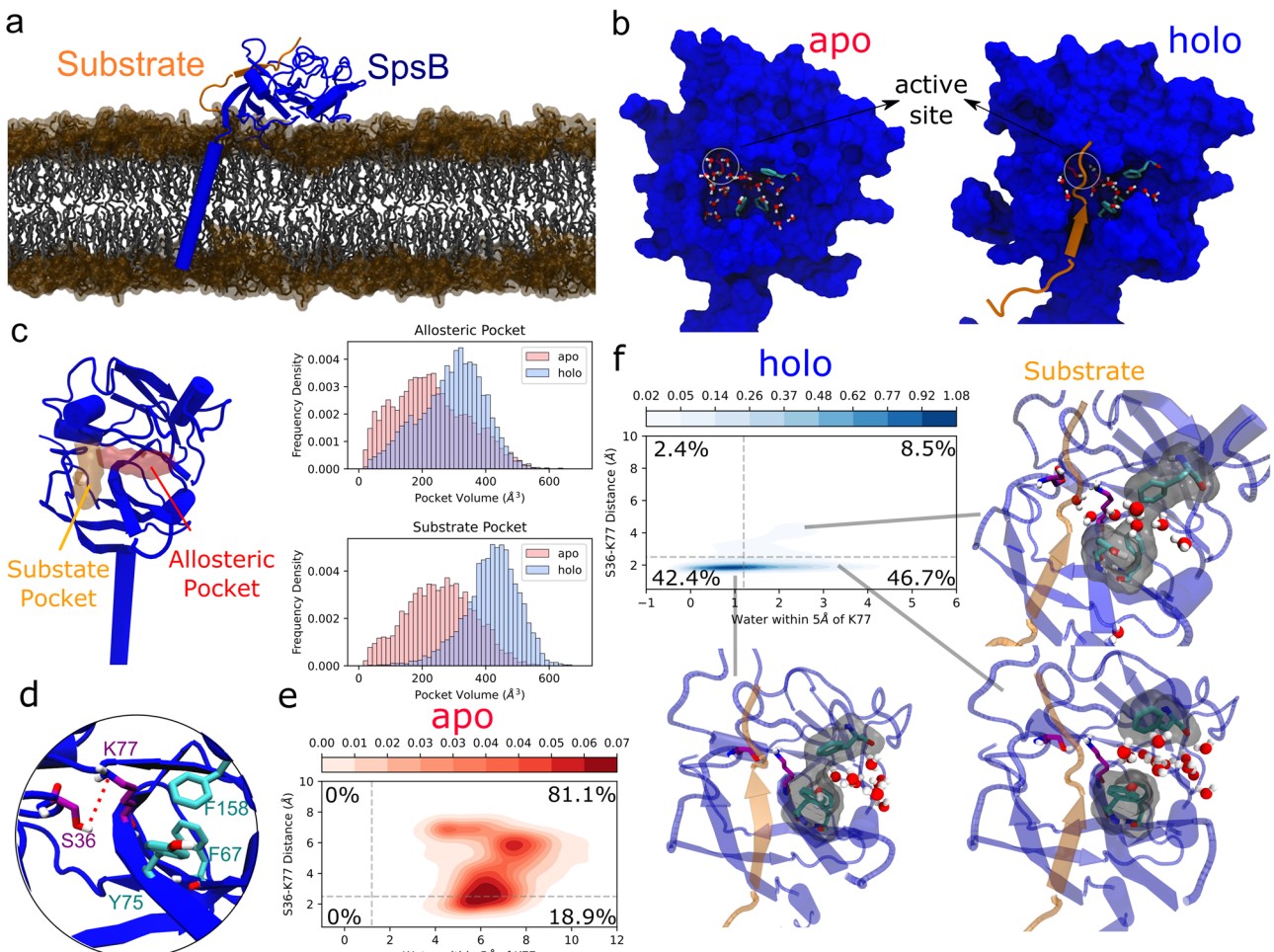

**Fig. 1 | Simulations of SpsB in lipid bilayer and the dynamics of the substrate and allosteric pocket. a** Schematic of the simulation system of interest. SpsB (blue) bound with the substrate (orange) is embedded in the POPC membrane bilayer. DMPG head groups are indicated as surface and the tails are shown in stick representation. **b** Distribution of the water molecules around the catalytic center. Water molecules with at least one atom within 5 Å of S36, K77, and F67 are shown. Residues F67, Y75, and F158 are depicted in the cyan stick representation. **c** Schematic of the substrate pocket (orange), the allosteric pocket (red), and their volume distribution in the apo (red) and holo (blue) simulations. **d** Geometry of the catalytic dyad S36 and K77 (magenta), and the neighboring aromatic residues F67, Y75, and F158 (cyan). The measured S36-K77 distance is highlighted by the red dashed line. **e** Distribution density of the catalytic S36-K77 distance and the number of water molecules around K77 in the apo simulation. **f** Distribution density of the S36-K77 distance and the number of water molecules around K77 in the holo simulation and its respective geometries near the catalytic center. The water count and S36-K77 distance in **e** and **f** are averaged over every 5 ns. The percentages in **e** and **f** represent the conformation populations, using a water count cut-off of 1.2 and a S36-K77 cut-off distance of 2.5 Å.

## Experimental assays validate critical residues for the activation mechanism of PK150

To experimentally validate the computational findings, single point mutations (F67A, Y75A, F158A, Q165A), double point mutations (F67A-Y75A, F67A-F158A, Y75A-F158A) and triple point mutations 3A (F67A-Y75A-F158A) were introduced and further subjected to a previously validated fluorescence resonance energy transfer (FRET) based SpsB activity assay in membrane fractions[20,36,37]. For this, *E. coli* membranes induced for SpsB expression (wildtype or respective mutant) were incubated with a SpsB peptide substrate equipped with a FRET donor/acceptor pair (Fig. 2e)[20]. As expected, PK150 enhanced the substrate turnover reproducibly[20] with a 1.7-fold elevated activity in wildtype SpsB (wt) at 10 µM (Fig. 2f). To further validate that this activation relied on specific binding to SpsB and not by general protein stabilization through e.g. compound aggregation[38], three detergents were added to the SpsB activity assay below their critical micellar concentration (0.01% Tween[39], 0.001% NP-40[40,41], 0.1% CHAPS[42–44] <cmc). Activation of wt SpsB with PK150 could still be shown for all detergents excluding unspecific assay interference effects (Supplementary Fig. S6)[38].

A similar enzyme overactivation was also observed for the Q165A mutant suggesting an insignificant role of Q165A in the activation

mechanism. Most importantly, the introduction of F67A, Y75A, and F158A mutations retained catalytic activity but lead to a loss of enzyme activation demonstrating their essentiality for PK150 interaction. Thus, these residues in the allosteric pocket are crucial for an enzyme activation by PK150 as predicted by MD simulations (Fig. 2f).

## PK150 activation of SpsB relies on interaction with critical residues F67, Y75, and F158A

Surprisingly, an unexpected small inhibitory effect of PK150 was observed for the single mutants F67A, Y75A, and F158A (20% inhibition) and moderate inhibition for F67A-Y75A, F67A-F158A, Y75A-F158A, 3A (30–40% inhibition) at 10 µM compared to DMSO in induced *E. coli* membranes (Fig. 2f). Of note, the expected activation curve of PK150 in wt reverses to an inhibition curve of PK150 in F158A mutant showcasing the change in regulatory mechanism of PK150 when crucial residues are mutated (Fig. 3a). To investigate how PK150 might inhibit SpsB mutants, we compared PK150 to a known active site SpsB inhibitor arylomycin A4[45] in wt SpsB and F158A mutant. Arylomycin A4 inhibits wt SpsB and F158A with low IC$_{50}$ values (210 nM for wt and 229 nM for F158A mutant) showing that the catalytic active site is generally not altered in the F158A mutant.

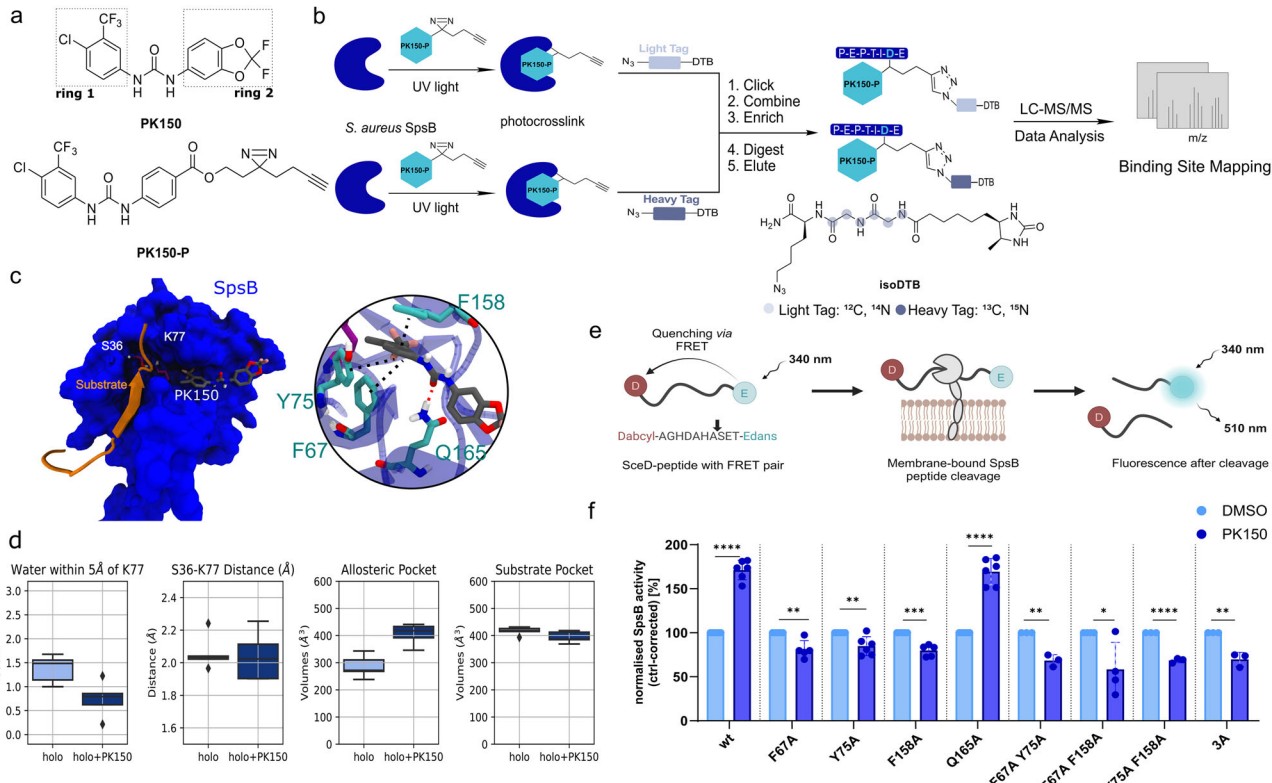

**Fig. 2 | Influence of the enzyme activator PK150 on SpsB. a** Chemical structures of PK150 (with indicated ring 1 and ring 2 and its photo crosslinker derivative PK150-P used in chemoproteomic workflow depicted in 3b. **b** Illustrated binding site identification workflow using isoDTB tags (isotopically labeled desthiobiotin azide) chemo proteomic workflow[29,30]. SpsB is incubated with PK150-P, irradiated with UV light for crosslinking. After Click chemistry, enrichment and further sample preparation, modified peptides are analyzed using LC-MS/MS and MSFragger-based FragPipe computational platform for data evaluation and binding site mapping[31–35]. **c** Scheme of the starting structure (pose 1) of the PK150-bound holo structure (holo +PK150). The π-π stackings between PK150 ring 1 and binding pocket are indicated by the black dashed lines and the hydrogen bond between PK150 and Q165 is indicated by the red dashed line. **d** Dynamics of holo-SpsB with and without PK150. From left to right: sampled distribution of the number of water molecules near K77, the S36-K77 distance, volume of the allosteric pocket, and volume of the substrate pocket in the holo (light blue), and holo+PK150 (dark blue) systems. Data points with PK150 RMSD > 15 Å are excluded from the analysis. Data points in the box plot are the mean of each simulation (n = 5). **e** Schematic representation of the FRET-based membrane-bound SpsB activity assay adapted from Le et al.[20]. The signal

peptide sequence of the *S. epidermidis* SceD preprotein modified with the FRET fluorophore EDANS (5-((2-aminoethyl)amino)-1-naphthalene sulfonic acid and the quencher DABCYL (4-((4-(dimethylamino)-phenyl)azo) benzoic acid is used as SpsB substrate. **f** Results of SpsB activity assay: PK150-induced (10 μM) cleavage of the FRET substrate by membrane-bound wt SpsB or respective mutants (50 μg ml⁻¹ total membrane protein concentration). The substrate cleavage rates are normalized to DMSO-treated samples from the induced membranes as quantification of SpsB in each membrane cannot be performed. Membranes were extracted from *E. coli* BL21(DE3)pLysS cells that harbor pET-55-DEST-full-length-SpsB or respective mutant. Non-induced *E. coli* membranes were subtracted as the baseline to account for potential background activity of the membrane. The bars highlight the SpsB activity at a PK150 concentration of 10 μM relative to DMSO. Data shown represent mean values ± s.d. of averaged triplicates of at least n = 3 biologically independent experiments per group (n = 3 for F67A-Y75A, Y75A-F158A, 3A, n = 4 for F67A-F158A, n = 5 for F67A, F158A, n = 6 for wt, Y75A, Q165A); P values: >0.05 (ns), <0.05 (*), <0.011 (**), <0.001 (***), <0.0001 (****). Two-tailed Student's t-test of compound- versus DMSO-treated groups was performed for each mutant for statistical significance.

To further investigate the molecular mechanism of the observed inhibition of the SpsB mutants, we performed additional MD simulations of all possible enzyme-substrate-ligand combinations (apo, apo+PK150, holo, holo+PK150) of wt and mutated SpsB. In PK150-free simulations (apo and holo), the allosteric and substrate pocket exhibited a less pronounced increase in volume upon substrate association in the holo F67A, F75A, and 3A systems, while the allosteric pocket of F158A occupies already an increased volume in its apo-form (Fig. 3b). Compared to the wt SpsB, the catalytic center is more hydrated in the apo+PK150 and holo simulations of the Y75A, F158A, and 3A mutants, albeit to a lesser extent in the F67A mutant, supporting the hypothesized water-gating roles of these residues (Supplementary Fig. S7). A correlation between active site water accessibility and S36-K77 distance is observed across different systems with a Pearson's R of 0.87, Kendull's of 0.69, and Spearman's R of 0.88. Notably, the trend of water accessibility follows apo > apo + PK150 > holo > holo + PK150, suggesting that the substrate and PK150 together dehydrate the SpsB active site (Supplementary Fig. S8).

Analysis on the dynamics of PK150 in the apo- and holo-enzymes showed dissociation events of PK150 in two out of five simulations of the wt apo SpsB (the dissociation rate is lower than observed for the holo enzyme, Fig. 3c) and the mutations tend to indicate slightly longer residence times for binding to apo SpsB compared to wt (Fig. 3c). Notably, if PK150 binds the apo SpsB, it can sample binding positions that partially overlapped with the active site region of SpsB (Fig. 3d) which was not observed in case of PK150 binding to the SpsB holo-form (Fig. 3e). Thus, binding of PK150 to the apo form is not favorable for enzyme activation in general since PK150 can possibly interfere with the catalytic center and may perturb substrate binding. Hence, we hypothesize that the inhibition effect of PK150 in SpsB mutations results from residue-specific deviations from the enzyme activation mechanism. According to our data, for activation to occur, PK150 binds to the wildtype holo SpsB (apo less favored, see Fig. 3c) with an allosteric pocket size at around 300–500 Å³ to decrease the water accessibility to the active site (Fig. 2d). However, in MD simulations, for F67A, Y75A, and 3A mutants, allosteric pocket expansion was not observed in holo SpsB, blocking the activation of SpsB (Figs. 3b and 4, upper pathway).

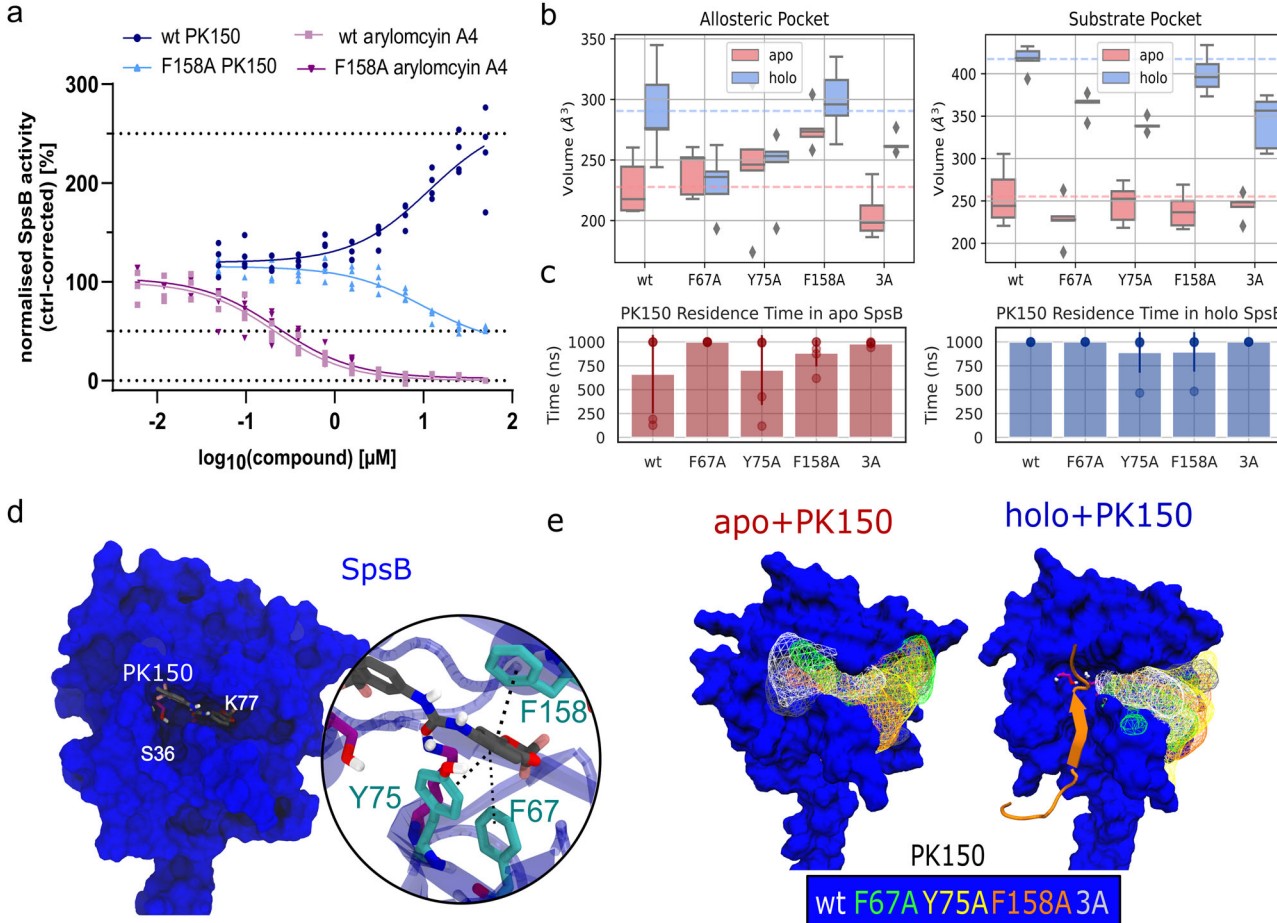

**Fig. 3 | Influence of mutations F67A, Y75A, F158A, and F67A-Y75A-F158A (3A) on SpsB. a** $IC_{50}$ and $EC_{50}$ curves: Normalized cleavage rates of the FRET substrate by membrane-bound wt SpsB or F158A mutant (50 μg ml$^{-1}$ total membrane protein concentration) with varying concentrations of PK150 or active site SpsB inhibitor arylomycin A4. The substrate cleavage rates are normalized to DMSO-treated samples from the induced membranes. Membranes were extracted from *E. coli* BL21(DE3)pLysS cells that harbor full-length SpsB or respective mutants. Respective non-induced membranes were subtracted as baseline. Data shown represent mean values ± s.d. of averaged triplicates of *n* = 3 biologically independent experiments per group (*n* = 4 for wt). Concentrations were log$_{10}$ transformed and $IC_{50}/EC_{50}$ curves were fitted with GraphPad Prism 10 using dose-response curves log (inhibitor/activator) with normalized responses and variable slope. $IC_{50}$ (arylomcyin A4) = 210 nM for wt and 229 nM for F158A mutant. Activation curve of PK150 in wt reverses to an inhibition curve of PK150 in F158A mutant showcasing

the change in regulatory mechanism of PK150 when crucial residues are mutated. **b** Volume of the allosteric pocket (left) and the substrate pocket (right) in the apo (light red) and holo (light blue) systems. Dashed lines indicate the corresponding average quantity from the wt systems. The circles and error bars show the average values ± s.d of the mean (*n* = 5). **c** Residence time of PK150 with ligand root-mean-square deviation less than 15 Å from the starting position. Data points indicate the residence time of PK150 in each simulation and the bar plot shows averaged values ± s.d (*n* = 5). **d** Schematic of the deviated binding mode of PK150 inside the allosteric pocket of wt apo-SpsB. The zoom-in view shows the π-π stackings between PK150 ring 2 and binding pocket (**e**) Sampled density of PK150 in the apo-enzyme (left) and holo-enzyme (right). The grid point sampled by PK150 with a frequency >10% of the simulation time is shown in the wireframe representation colored based on the mutation of SpsB (wt in white, F67 in green, Y75 in yellow, F158A in orange, and 3A in gray).

For F158A, the increased size of the allosteric pocket enhances the likelihood of PK150 binding to the apo-form with a deviated binding mode, perturbing substrate binding (Figs. 3b and 4, lower pathway).

## Discussion

SpsB controls a key enzymatic step in protein secretion that enables mature protein release after translocation in the membrane[22,24]. Contrary to well-explored inhibitory pathways, Le et al. recently published a novel antibiotic compound PK150 that stimulates the activity of SpsB and linked its contribution to killing MRSA[20]. However, enzyme dynamics and a precise activation mechanism of PK150 were so far not fully understood, partly because the binding site was not known. We therefore experimentally identified the PK150 binding site by an isoDTB-based chemical proteomics workflow and in addition deciphered a substrate and an allosteric pocket close to the active site dyad S36 and K77 by MD simulations. These two pockets recruit water molecules to the catalytic dyad, which is crucial for catalyzing the peptide cleavage. From a conformational point of view, the

motion of the residues F67A, Y75A, and F158 relative to each other controls the water access to the active site and impairs the hydrogen bond between S36 and K77 that is needed in the first step of catalysis. Access of water to the active site plays a pivotal role in the catalysis mechanism, needed in the ultimate step to release the acylated signal peptide from the enzyme[22,24,27]. However, an intact hydrogen bond between S36 and K77 is crucial in the first step of catalysis, in which water molecules can act as competitors for hydrogen bonding to S36 and K77 and have an important impact on the strength of this hydrogen bond and the corresponding rate of catalysis[22,27]. In line with our previous studies on an intramembrane peptidase, the number of water molecules around the active site correlates with the stability of the catalytic geometry of SpsB in the active site[46].

Based on chemical proteomics studies and MD simulations, we suggest that binding of PK150 into the allosteric pocket, which is expanded in size upon substrate association, reduces the accessibility of water to the active site. Consequently, the distance between S36 and K77 is reduced, and catalysis is enhanced. In addition, the MD simulations suggested central

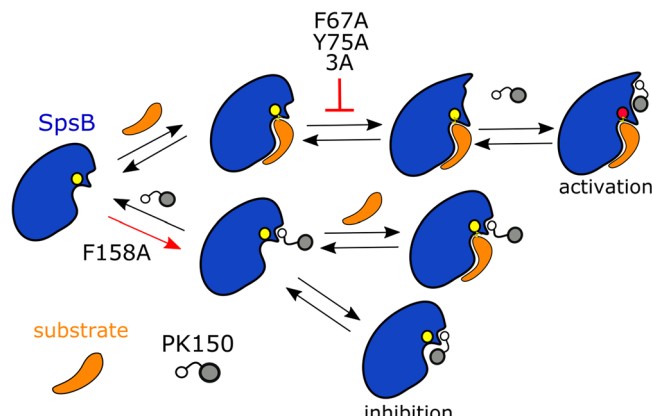

**Fig. 4 | Putative allosteric activation and inhibition pathways of PK150 in SpsB.** PK150 promotes the enzyme activity when it binds to the holo-enzyme but inhibits the activity when it partially occupies the substrate pocket (lower pocket) of the apo-SpsB. Upper pathway: expansion of the allosteric pocket (upper pocket) via substrate binding facilitates the recruitment of PK150 and leads to the activating regulation which is mainly observed in wt and Q165A mutant. Lower pathway: binding of PK150 to the apo enzyme leads to a non-competitive inhibitory regulation. Here, mutations F67A, Y75A, and 3A diminish the expansion of the allosteric pocket upon substrate binding (red opposite T shape), and mutation F158A promotes the binding of PK150 to the apo-pocket (red arrow) leading to an overall inhibition effect upon introduction of F67A, Y75 or F158A mutations. SpsB is shown in blue, PK150 in gray, and substrate in orange. The catalytic center is shown with normal (yellow) or enhanced (red) activity.

interactions between F67, Y75, F158, Q165 and PK150. We experimentally investigated these predictions via a FRET-based SpsB activity assay of wt SpsB and mutants in *E. coli* membranes. PK150 enhanced the substrate turnover with a 1.7-fold elevated activity in wt SpsB and Q165A mutant showing an insignificant role of Q165A hydrogen bonding with the urea moiety of PK150 on the activation mechanism. However, the introduction of either single, double, or triple F67A, Y75A, and F158A mutations leads to a general loss of the activation effect underlining that all three predicted residues are crucial for SpsB activation by PK150. These results indicate that the overall mechanism of activation is based on hindering the access of water into the active site by interaction of F67A, Y75A, and F158A with PK150 in the allosteric pocket of SpsB. This reduced water influx results in a stronger H-bond formation between the active site residues S36 and K77, promoting the catalysis rate of SpsB.

If one of the interactions between PK150 and these residues in the allosteric pocket is disturbed e.g. via introduction of mutations, the PK150 activation mechanism reverses to a small to moderate inhibitory effect. MD data suggests different regulatory pathways depending on whether PK150 binds to the apo or holo SpsB enzyme (Fig. 4f). The expansion of the allosteric pocket upon substrate binding (holo) facilitates the recruitment of PK150 and leads to the activating mechanism by limiting the water accessibility to the active site which is mainly observed for PK150 in wt SpsB. However, binding of PK150 to the apo enzyme leads to a deviated binding mode interfering with the active site region in a non-competitive inhibitory manner. We suggest that the overall inhibitory effect may result from either diminished allosteric pocket expansions in holo SpsB (for F67A, Y75A and 3A mutants) blocking the activation or the increased size of the allosteric pocket enhances the likelihood of PK150 binding to the apo-form favoring the apo form of SpsB (F158A) leading to perturbations of substrate binding. This showcases most importantly the essentiality of these residues by interacting with PK150 in the allosteric pocket to block the water access to the catalytic center enabling enzyme activation of SpsB by PK150 as predicted by MD simulations.

In summary, this study is the first to demonstrate a strategy for enzyme activation that uses a small molecule to control water accessibility to the active site, thereby enhancing the catalysis rate. We additionally observed

that the activation mechanism can turn into an undesired inhibitory regulation if the allosteric binding pocket is altered e.g. by the introduction of mutations of these crucial residues, which then leads to decreased pocket volumes or interferences of PK150 with the substrate binding site. These findings highlight the importance of combining computational and experimental techniques to identify crucial amino acid residues that are responsible for compound stabilization in the allosteric pocket. As an outlook, this study opens the possibility of designing activators with enhanced binding affinity, either by increasing interaction strength with known critical residues F67, Y75, and F158 or by establishing novel interactions with residues in the allosteric pocket to reduce water influx into the active site.

## Methods
### MD simulation and analysis
The complete structure of the substrate-bound SpsB is constructed by manually fusing I26 to K193 of SpsB taken from PDBID 4WVJ[21] and M1 to F25 predicted by Alphafold2[47]. The substrate (GGGGGAPTAKAA*SK, * is the scissile bond) is mutated from the peptide inhibitor taken from PDBID 4WVJ. A substrate-free SpsB structure is generated by removing the substrate from the modeled substrate-bound SpsB. In silico mutation on SpsB and the substrate are introduced by tleap from Ambertools 18[48]. All systems are embedded in the dimyristoyl phosphatidylglycerol (DMPG) bilayer to mimic the bacteria membrane[49] and solvated in water and 0.15 M NaCl using CHARMM-GUI online server[50]. Interaction between atoms is described by ff19SB[51] for proteins, lipid21[52] for membrane, OPC[53] for the water molecules and GAFF2[54] for PK150. K77, with a predicted pKa value of 7.48 by propKa3[55] is unprotonated for the formation of the initial catalytic geometry. The total potential energy of each simulation box was minimized with maximal 70,000 steps with 10 kcal·mol$^{-1}$·Å$^{-2}$ positional restraint on protein using the MPI version of the pmemd program in Amber22, followed by equilibration with gradually releasing positional restraint, from 10 to 0.1 kcal·mol$^{-1}$·Å$^{-2}$ on protein and 2.5 to 0 kcal·mol$^{-1}$·Å$^{-2}$ on the membrane, for 400 ps at 303.15 K using the cuda version of pmemd and a non-bonded cutoff distance of 9 Å[56]. The equilibrated systems are submitted for 1 μs production run at a temperature of 303.15 K using the Langevin thermostat[57] and a pressure of 1 bar by Berendsen barostat[58]. A time step of 4 fs was allowed with the SHAKE algorithm[59] and hydrogen mass repartitioning method[60]. The allosteric pocket and the substrate pocket were first identified using MDpocket[28] on the apo-form SpsB simulations using the default pocket parameters where the coordinates of the probes for each pocket were extracted. The size of the allosteric pocket and substrate pocket were calculated based on the pocket formation criteria using the "pocket selection" function with the default fpocket parameter set in MDpocket[28]. Water accessibilities and geometric measurements between the atoms are calculated using PYTRAJ and CPPTRAJ[61]. Initial docking mode of PK150 is selected from the docking poses generated by docking PK150 to a snapshot of holo-SpsB where the allosteric pocket is in an open form using AutoDock Vina[62] according to the description in the main text. The point charges of PK150 are assigned according to the AM1-BCC charge model[63] using antechamber[64]. Time evolution of the root-mean-square deviation (RMSD) of enzymes, substrate, and ligands are shown in Supplementary Figs. S9–S11. Time evolution of the volume of the allosteric pockets and substrate pockets are shown in Supplementary Figs. S12 and S13. Time evolution of the S36-K77 distance and the number of water molecules within 5 Å is shown in Supplementary Figs. S14 and S15. Root-mean-square deviation of the enzymes and substrates are shown in Supplementary Figs. S16 and S17, respectively.

### Chemical compounds
PK150 ((4-Chloro-3-(trifluoromethyl)phenyl)-3-(2,2-difluorobenzo[d][1,3]dioxol-5-yl)urea) and PK150-P(2-(3-(But-3-inyl)-3H-diazirin-3-yl) ethyl4-(3-(4-Chloro-3trifluoromethylphenyl)ureido) benzoate) were previously synthesized according to a published procedure by our group[20]. Heavy and light isoDTB (isotopically labeled desthiobiotin) tags were kindly provided by Dr. Stephan Hacker (Leiden University)[29]. Arylomycin A4 was

a kind gift from Prof. Dr. Stephanie Grond (Eberhard Karls Universität Tübingen). All other chemical compounds used within this work were commercially available and used without further purification.

## Bacterial strains

*E. coli* strains used within this work (BL21(DE3), BL21(DE3)pLysS and XL1Blue) were cultivated in LB medium (Lysogeny Broth; 10 g/L casein peptone, 5 g/L NaCl, 5 g/L yeast extract, pH 7.5) and supplemented with respective antibiotics if indicated.

## Plasmids

Plasmids and their characteristics used for the methods in the following sections including protein purification of MBP-tagged SpsB, site-directed mutagenesis of full-length SpsB, and generation of *E. coli* membranes with full-length SpsB (wt and mutants) are summarized in Supplementary Table S1.

## Recombinant expression and purification of MBP-tagged *S. aureus* SpsB from *E. coli*

For protein expression, the vector pETMBP-1a-His-MBP-SpsB was transformed in *E. coli* BL21 (DE3). Cells obtained from an overnight culture in LB Medium (supplemented with 50 µg/ml Kanamycin) were diluted 1:100 in LB medium (50 µg/ml kanamycin) and incubated at 37 °C (180 rpm) until an $OD_{600}$ of 0.45–0.6 was reached. Protein expression was induced with 0.3 mM isopropyl-β-D-1-thiogalactopyranoside (IPTG) and incubated for 3 h (25 °C, 180 rpm). Cells were subsequently harvested by centrifugation (5000 × g, 10 min, 4 °C). All following purification steps were performed at 4–8 °C or on ice unless otherwise noted. A two-step purification procedure of MBP-SpsB protein was applied comprising MBP-affinity chromatography followed by a size-exclusion chromatography (SEC), both carried out on an ÄKTA-FPLC system (*GE Healthcare*, now *Cytiva*). The cell pellet (obtained from 1 L expression culture) was resuspended in 30 mL lysis buffer (20 mM Tris, pH 8, 200 mM NaCl, 1 mM EDTA, 5% (w/v) glycerol, 1 mM dithiotreitol (DTT)) supplemented with 0.1 mg/ml DNase I (*AppliChem*) and one tablet cOmplete TM ULTRA EDTA-free protease inhibitor tablet (*Roche*). Cells were lysed by homogenization using an EmulsiFlexC5 (*Avestin Inc.)* and cleared lysate was obtained by centrifugation (24,446 × g, 30 min, 4 °C) and filtration using a Whatman TM folded filter (*GE Healthcare*, now *Cytiva*). The cleared lysate was loaded onto a MBPTrap HP column (*Cytiva*). After sample application, the column was washed with wash buffer (20 mM Tris, pH 8, 200 mM NaCl, 1 mM EDTA, 5% (v/v) glycerol, 1 mM DTT), followed by elution of MBP-SpsB with elution buffer (20 mM Tris pH 8, 200 mM NaCl, 1 mM EDTA, 5% (v/v) glycerol, 1 mM DTT, 10 mM maltose). Fractions containing the protein were pooled and concentrated with an Amicon Ultracell Centrifugal filter unit (MWCO 10 kDa, *Merck Millipore*) and applied onto a HiLoad Superdex 75 (16/60) column (*Cytiva*), equilibrated with SEC buffer (20 mM Tris-HCl pH 8, 200 mM NaCl, 5% (v/v) glycerol, 1 mM DTT). Fractions containing the pure protein were pooled, concentrated, and aliquots snap-frozen in liquid nitrogen and stored at −80 °C. Protein purity was confirmed by SDS-Page (Supplementary Fig. S4, molecular weight: 62033.20 Da)

## Binding site identification using isoDTB chemical proteomics workflow

A mass spectrometry workflow using isotopically labeled desthiobiotin azide tags (isoDTB workflow) according to a published procedure[29,30] was performed to determine the binding site of photocrosslinker PK150-P to SpsB.

Four samples of recombinantly purified MBP-tagged extracellular domain of SpsB protein (see Experimental procedure 'Recombinant Expression and Purification of MBP-tagged *S. aureus* SpsB from *E. coli*') were diluted to a final concentration of 20 µM in PBS (final volume: 90.5 µL per well in 96-well-plate). A photocrosslinker derivative of PK150, PK150-P was added to each sample (final concentration: 20 µM) and incubated for 1 h at room temperature. For photocrosslinking of PK150-P to SpsB, samples were irradiated with UV light (Hitachi FL8BL-B lamps) for 10 min and then transferred to Protein LoBind tubes (*Eppendorf*).

A Click mix solution with heavy and light isoDTB (desthiobiotin azide) azide tags was prepared, where in total two tubes with TBTA (Tris[(1-benzyl-1H-1,2,3-triazol-4-yl)methyl]amin, 18 µL, 0.9 mg/mL stock in 4:1 *t*BuOH/DMSO), TCEP (Tris(2-carboxyethyl)phosphine, 6 µL, 13 mg/mL stock in ddH₂0) and CuSO₄ (6 µL, 50 mM stock in ddH₂0) were prepared. 6 µl of heavy or light isoDTB tag (5 mM stock in DMSO) was added to one tube. Click mix solution (12 µL) was added to the SpsB samples (2 × heavy, 2 × light) and incubated at room temperature for one hour. After combining heavy and light samples of each biological replicate, cold acetone (800 µL, −20 °C) was added to both tubes, and samples were incubated at −20 °C for at least 2 h for full protein precipitation. The samples were centrifuged (13,000 × g, 4 °C, 10 min) and the resulting protein pellet was washed two times with MeOH (500 µL, −20 °C) *via* mild sonication (10%, 5 × cycle, 10 s) and centrifugation (13,000 × g, 4 °C, 10 min). The supernatant was removed, and the protein pellet was air-dried for 10 min. After dissolving the pellet in urea (60 µl, 8 M in 0.1 M TEAB, triethylammonium bicarbonate buffer) *via* mild sonication (10%, 5 × cycle, 10 s), the sample was centrifuged (13000 × g, r.t., 3 min). For reduction of disulfide bonds, DTT (dithiothreitol, 3 µL, 31 mg/ml in ddH₂O) was added to the samples and incubated at 37 °C for 45 min with constant shaking (850 rpm, Thermomixer, *Eppendorf*). IAA (iodoacetamide 3 µL, 74 mg/mL stock in ddH₂O) was added for cysteine carbamidomethylation and incubated at 37 °C for 30 min while shaking (850 rpm, Thermomixer, *Eppendorf*). After addition of DTT (3 µl, 31 mg/ml stock in ddH₂O) to quench excess IAA, samples were shaken at 37 °C for further 30 min (850 rpm, Thermomixer, *Eppendorf*). 180 µl of 0.1 M TEAB was added to each tube and samples were digested with 4 µl of trypsin (ratio trypsin/protein 1:100, 0.5 µg/µL, *Promega*) overnight while shaking (37 °C, 220 rpm, Incubator Shaker, *Eppendorf New Brunswick*).

Next day, high-capacity streptavidin agarose beads (*Fisher Scientific*) were prepared: Beads (25 µL initial slurry per sample, 50 µL in total) were washed three times with NP-40 substitute (1 mL, 0.1% in PBS, 3000 × g, r.t., 3 min). The supernatant was aspirated, and NP-40 substitute (1.2 mL, 0.2% in PBS) was added to the beads. 600 µl of beads were added to each digested peptide sample and incubated for at least 1 h at r.t. while rotating (disc rotator) to enrich peptides containing isoDTB tags on the beads. Samples were centrifuged (1000 × g, r.t., 2 min) and the supernatant was removed. NP-40 (600 µL, 0.1% in PBS) was added to each sample and transferred to centrifugation columns (*Fisher Scientific* Pierce™). Each sample was washed with NP-40 (1 × 600 µL, 0.1% in PBS), with PBS (3 × 600 µL, and ddH₂O (3 × 600 µL). Peptides were eluted into Protein LoBind tubes (*Eppendorf*) with 1 × 200 µL and 2 × 100 µL trifluoroacetic acid (TFA, 0.1% in 50% aqueous MeCN) followed by final centrifugation (3000 × g, 3 min, r.t.). The solvent was removed in a vacuum concentrator (5 h, 30 °C, Concentrator Plus, *Eppendorf*) and the resulting dried peptides were stored at ˗20 °C until analysis.

The samples were analyzed according to a published procedure[30].

Before mass spectrometry dried peptides were reconstituted in 0.1% (v/v) TFA *via* sonication bath (3 × 5 min, *Bandelin* Sonorex) and filtered through 0.22 µm PVDF filters (*Millipore*). Samples were transferred to MS vials and analyzed *via* HPLC-MS/MS using an UltiMate 3000 nano HPLC system (*Dionex*) equipped with Acclaim C18 PepMap100 trap column (75 µm ID × 2 cm, Acclaim, *ThermoFisher*) and Aurora Ultimate™ (3ʳᵈ generation, 20 cm nanoflow UHPLC compatible, *ionopticks*) separation columns coupled to a Q Exactive Plus Orbitrap Mass Spectrometer (*Thermo Fisher Scientific*). Samples were loaded onto the trap column and washed with TFA (0.1% in ddH₂O). The subsequent separation was carried out with a flow rate of 400 nL/min using buffer A (0.1% formic acid (FA) in water) and buffer B (0.1% FA in acetonitrile). The separation column was heated to 40 °C. The analysis started with washing for 7 min with 5% buffer B for desalting followed by a gradient from 5% to 40% buffer B over 105 min, a second gradient from 40% B to 60% B within 10 min, and a final increase to 90% B in 10 min. Isocratic washing with 90% B was performed for 10 min, then decreased to 5%

in 0.1 min and held at 5% for additional 9.9 min for re-equilibration. The Q Exactive Plus mass spectrometer was run in a TOP10 data-dependent mode. In the orbitrap, full MS scans were collected in a scan range of 300–1500 m/z at a resolution of 70,000 and an AGC target of 3e6 with 80 ms maximum injection time. The TOP10 peaks were selected for MS2 scan with a minimum AGC target of 1e3 and isotope exclusion and dynamic exclusion (exclusion duration: 60 s) enabled. Peaks with unassigned charges or a charge of +1 were excluded. Peptide match was "preferred". MS2 spectra were collected at a resolution of 17,500 aiming at an AGC target of 1e5 with a maximum injection time of 100 ms. Isolation was conducted in the quadrupole using a window of 1.6 m/z. Fragments were generated using higher-energy collision-induced dissociation (HCD, normalized collision energy: 27%) and finally detected in the orbitrap.

## Data analysis of LC-MS/MS spectra
The data analysis described in the following sections was performed according to a published procedure[30] and adjusted for binding site identification studies. Obtained tables and filtered results after performing open and modified closed searches are summarized in the Supplementary Data S1.

Acquired raw data from performed LC-MS/MS analyses was converted into a mzML format using the MSconvert tool (version: 3.0.21193-ccb3e0136) of the ProteoWizard software (version: 3.0.21193 64 bit)[65] using standard settings with vendor's peak picking enabled. For further data analysis, the FragPipe interface (version: 20.0) with MSFragger (version 3.8)[31–35], Philosopher (version: 5.0.0)[66], IonQuant (version 1.9.8)[67] and Python (version 3.7.3) was used. A FASTA database for *S. aureus* NCTC8325 was downloaded from www.uniprot.org on 22.08.2023[68]. The reverse sequences were manually added to the FASTA databases.

### *Open Search Analysis* of mass of modifications with FragPipe[31–35,66,67]
To survey the landscape of all mass shifts observed on the peptides of MBP-SpsB in the data set, an *Open Search* was performed with MSFragger[31–35,66,67]. For this purpose, the following settings were used: Precursor mass tolerance −150 to 1000 Da, (initial) fragment mass tolerance 20 ppm, Calibration and Optimization 'Mass calibration, parameter optimization' enabled, Isotope Error '0', enzyme name 'trypsin', cut after 'KR', but not before 'P', cleavage 'enzymatic', missed cleavages '2', Clip N-term N enabled, peptide length 7–50, peptide mass range 500–5000 Da, no variable modifications, no fixed modifications, all other options were left at the standard settings. Crystal-C[33] was enabled. PeptideProphet[66] was run with the following setting: '--non-param --expectscore --decoyprobs --masswidth 1000.0 --clevel -2'. PTMProphet was disabled. ProteinProphet[66] was run with the following settings: '--maxppmdiff 2000000'. Generate report was enabled with the following settings: '--sequential --razor –mapmods --prot 0.01'. Run MS1 quant was disabled. Run TMT-Integrator was disabled. PTM-Shepherd[34] was enabled with the following settings: Smoothing factor '2', Precursor tolerance '0.01 Da', Prominence ratio '0.3', Peak picking width '0.002 Da', Localization background '4'. Annotation tolerance '0.01 Da', Custom mass shifts: a custom mass shift list was used including only UniMod modifications with less than 400 Da molecular weight as previously published[30], Ion Types for modification with 'b' and 'y' enabled and mass fragment charge '2'. Generate Spectral Library was disabled. For downstream data analysis, the 'global.modsummary.tsv' file was searched for mass shifts >482 Da (Exact mass of LightTag: 482.2834 Da, HeavyTag: 488.2909 Da) and with mass shift differences of 6.0075 ± 0.0010 Da Da between heavy and light isoDTB tags clicked to PK150-P. Filtered results are summarized in Supplementary Table S2.

### Modified *Closed Search Analysis* for binding site identification studies
To identify the binding site of PK150-P a modified *Closed Search* was performed. For this purpose, the following settings were used in MSFragger[31–35,66,67]. Precursor mass tolerance −20 to 20 ppm, fragment mass

tolerance 20 ppm, Calibration and Optimization 'mass calibration, parameter optimization", Isotope error '0/1/2', enzyme name 'trypsin', cuter after 'K', but not before 'P', cleavage 'enzymatic', missed cleavages '2', Clip N-term N enabled, peptide length 7–50, peptide mass range 500–5000 Da, no mass offsets, all other options were kept at default settings. Fixed and Variable modifications were set to the detected masses 931.3724 Da and 937.3792 Da found in the OpenSearch run corresponding to the mass of PK150-P-isoDTB tag adduct Crystal-C was disabled. PeptideProphet was run with the following settings: '--decoyprobs --ppm --accmass --nonparam –expectscore'. PTMProphet was disabled. ProteinProphet was run with the following settings: '--maxppmdiff 2000000'. Generate report was enabled with the following settings: '--sequential --prot 0.01'. PTM-Shepherd was disabled. Run MS1 quant was enabled with the following settings: IonQuant enabled, Add MaxLFQ enabled with MaxLFQ min ions '2', labeling based quant with the detected masses indicated in Supplementary Table S2 on all amino acids (*), Re-quantify enabled, Top N ions '3', Min freq. '0.5', Min scans '1', Min isotopes '2', Normalize disabled. RT window (minutes) '0.4' and m/z Window (ppm) '10'. Run TMT-Integrator was disabled. Generate Spectral Library was disabled. For duplicates, both runs were analyzed as different experiments. For downstream data analysis, the 'ion_label_quant.tsv' files of the two experiments were analyzed separately. For each entry, the 'Modified peptide' was generated as either the 'Light Modified Peptid' or the 'Heavy Modified Peptide' based on the entry with the higher 'PeptideProphet Probability'. The masses of probe modification in the 'Modified Peptide' were replaced by an '*' and the mass of carbamidomethylation ([57.0215]) in this entry was deleted if present. The full protein sequence was linked to the table. Based on this information, all peptide sequences that do not occur exactly once in the same protein were excluded and the residue number of the modified residue was determined. The 'Identifier' was generated in the format 'UniProtCode'_*residue number, where * represents the one-letter code of the respective modified amino acid. For each 'identifier', the averaged 'Log2 Ratio HL', which is the log2 transformed ratio of heavy and light ions, was determined as the average of the 'Log2 Ratio HL' of all corresponding ions weighted with the 'Total intensity' of the ion, which was calculated as the sum of 'Light Intensity' and 'Heavy Intensity' for each ion. The value was disregarded if the standard deviation of the 'Log2 Ratio HL' values was >1.41 for all ions of the same 'Identifier'. Additionally, for each 'Identifier' the 'Total Intensity', 'Total Light Intensity' and 'Total Heavy Intensity' was calculated as the sum of all 'Total Intensity', 'Light Intensity' and 'Heavy Intensity' values of the individual ions, respectively. If several different 'Modified peptides' were detected for the same 'Identifier', the 'Modified Peptide' and the 'Peptide Sequence' with the shortest sequence were kept. For all identifiers, the data for both replicates were now combined in one table. The 'Log2Ratio HL' values for the replicates were named 'Log2 ratio HL replicate 1' and 'Log2 ratio HL replicate 2'. The average of these two values was calculated and named 'Log2 ratio HL'. The value was disregarded if the standard deviation between the replicates was >1.41 or if the identifier was only quantified in one of the replicates. Remaining peptide sequence that fulfilled all the mentioned criteria was assigned to be the binding site of PK150-P to the protein SpsB (Supplementary Table S3).

### QuikChange site-directed mutagenesis for generation of SpsB point mutations
Cloning of full-length SpsB (fl-SpsB) was performed using the Invitrogen Gateway Technology in previous works by Le et al. [20]. fl-SpsB was cloned into the destination vector pET-55-DEST (*Novagen*) and transformed into chemically competent *E. coli* BL21(DE3)pLysS cells (Promega) to prevent leaky protein overexpression. Plasmid DNA of wildtype SpsB pET-55-DEST-fl-SpsB was purified using a plasmid Miniprep Kit (peqGOLD Plasmid Miniprep Kit II, *VWR Peqlab*) according to the manufacturer's instructions and used as template for all single point mutations (F67A, Y75A, F158A, Q165A). For double (F67A-Y75A, F67A-F158A, Y75A-F158A) or triple point (F67A-Y75A-F158A) mutations, the respective pET-55-DEST-fl-SpsB single point mutation or double point mutation plasmids

were isolated and used as template for site-directed mutagenesis. Point mutation primers (*Sigma*) were designed based on the pET-55-dest-fl-SpsB construct according to the manufacturer's instructions (Agilent). Primer sequences and order of point mutation introduction for all point mutations are listed in Supplementary Table S4. QuikChange site-directed mutagenesis PCR reactions were performed using either Phusion High-Fidelity DNA Polymerase (*New England BioLabs*, mutations: F67A, Q165A) or Q5 High-Fidelity DNA Polymerase (*New England BioLabs*, mutations: Y75A, F158A, F67A-Y75A, F67A-F158A, Y75A-F158A, F67A-Y75A-F158A) using the cycle listed below (Supplementary Table S5). The PCR mixture contained 10 μL GC buffer (NEB), 1 μL dNTP mix (10 mM), 1 μL forward primer (10 μM), 1 μL reverse primer (10 μM), 1 μL plasmid template (25 – 50 ng), 1.5 μL DMSO and 1 μL Phusion or Q5 High Fidelity DNA polymerase (NEB) and 33 μL ddH$_2$O.

PCR product was digested with DpnI (1 μL CutSmart Buffer, 1 μL DpnI, 8 μL PCR reaction mixture) for 1 h at 37 °C and subsequently transformed in *E. coli* XL1 blue cells for nick repair. After re-isolation of plasmid DNA with plasmid Miniprep Kit (peqGOLD Plasmid Miniprep Kit II, *VWR* Peqlab) according to the manufacturer's instructions, correct insertion of the desired point mutation(s) and the overall sequence was checked with Sanger Sequencing (*Azenta*). Isolated plasmid DNA was re-transformed in competent *E. coli* BL21(DE3)pLysS cells(*Promega*) as the final expression strain.

### Preparation of *E. coli* membrane (control) fractions

The preparation of *E. coli* membrane fractions was performed as previously described by Therien et al. and Le et al.[20,37].

For preparation of *E. coli* membrane fractions harboring overexpressed *S. aureus* SpsB, *E. coli* BL21(DE3) pLysS cells harboring pET-55-DEST-fl-SpsB or respective mutant plasmids (F67A, Y75A, F158A, Q165A, F67A-Y75A, F67A-F158A, Y75A-F158A, F67A-Y75A-F158A) were grown in LB medium (supplemented with Ampicillin (100 μg/mL) and Chloramphenicol (34 μg/mL)) at 37 °C (200 rpm) to OD$_{600}$ ≈ 0.6 and protein overexpression was induced by the addition of isopropyl-1-thio-β-galactopyranoside (IPTG, 0.5 mM). After incubation (22 °C, 200 rpm, 3 h), cells were harvested by centrifugation (6000 × $g$, 4 °C, 10 min) and subsequently washed with PBS. The cell pellet was resuspended in 5 mL Tris-HCl buffer (50 mM, pH = 7.5) and lysed using a bead beater homogenizer (3 cycles: 5500 rpm for 45 s, 30 s cooling; Precellys Ceramic Kit CK01L, 7.0 mL tubes; Precellys 24 Homogenizer, Bertin Technologies). The lysate was centrifuged (12,000 × $g$, 4 °C, 10 min) to remove intact cells and debris. Membranes were collected from supernatant (39,000 × $g$, 4 °C, 75 min) and resuspended in ice-cold sodium phosphate buffer (50 mM, pH 7.5). Protein concentration was measured using the Pierce BCA Protein assay kit (Thermo Fisher Scientific, Pierce Biotechnology) and snap-frozen aliquots were stored at −80 °C.

For preparation of respective *E. coli* membrane control fractions without overexpressed SpsB, procedure above was performed identically without addition of IPTG.

### FRET-based signal peptidase assay with membrane-bound SpsB

Signal cleavage activities of recombinantly expressed full-length *S. aureus* SpsB (wt) or mutants (F67A, Y75A, F158A, Q165A, F67A-Y75A, F67A-F158A, Y75A-F158A, F67A-Y75A-F158A) in *E. coli* membranes were measured using a Förster Resonance Energy Transfer (FRET) assay as previously described[20,36].

A synthetic peptide substrate based on SceD modified by 4-(4-dimethylaminophenylazo)benzoic (DABCYL) acid and 5-((2-aminoethyl)amino)-1- naphthalenesulfonic acid (EDANS) was used (DABCYL-AGHDAHASET-EDANS, *AnaSpec*) as FRET substrate. Assays were performed with overexpressed wildtype SpsB (wt) or mutant (F67A, Y75A, F158A, Q165A, F67A-Y75A, F67A-F158A, Y75A-F158A, F67A-Y75A-F158A) in *E. coli* membranes. As controls for background activity, respective *E. coli* membranes without IPTG induction were used (see Experimental

procedure "Preparation of E. coli membrane (control) fractions. Membranes containing recombinant wildtype SpsB or mutant (50 μg/mL membrane concentration, 100 μL/well final volume, sodium phosphate buffer, 50 mM, pH 7.5, addition of detergent if indicated; final concentrations in buffer: 0.1% CHAPS, 0.001% NP-40 or 0.1% Tween) were pretreated with PK150, Arylomycin A4 or DMSO as a control (final concentration 1%) at 37 °C for 5 min. After addition of FRET substrate (10 μM final substrate concentration; final DMF concentration from substrate stock 1%), fluorescence turnover was monitored by an Infinite™ M Nano Tecan 200Pro reader (340 nm excitation and 510 nm emission wavelengths, fluorescence top reading mode) at 37 °C for at least 2 h.

### Data analysis and statistics of FRET assay

After subtraction of basal membrane activity of non-induced *E. coli* membranes (non-induced *E. coli* membranes (wildtype or mutant) that are not containing any SpsB were shown to have minor to no substrate cleavage activity and were included in each experiment as a control), initial substrate cleavage velocities were determined *via* simple linear regression (linear range, t = 1000 s, GraphPad Prism 9/10), normalized to DMSO-treated samples of induced *E. coli* membranes, and plotted against PK150.

The assay was performed in technical triplicates for at least three biologically independent experiments ($n \geq 3$). Significance of cleavage activity was determined *via* a two-tailed Students t-test for comparison of membranes treated with DMSO to PK150.

To circumvent a misleading data interpretation by the reader, normalized data to each DMSO control is separately shown for each mutant. Showing unnormalized data would result in a misleading interpretation of the data, as one might directly compare the substrate cleavage rates among the mutants, concluding that this directly correlates with the enzyme activity of each mutant, which would be incorrect since quantification of SpsB (wildtype or mutant) in each membrane cannot be performed.

### Reporting summary
Further information on research design is available in the Nature Portfolio Reporting Summary linked to this article.

## Data availability
All simulation data including input and output files and trajectory files are available upon request from the authors. Source data, initial PDB files, and the code for generating the figures are available at *Zenodo*[69]. Results of binding site data analysis can be found in the Supplementary Data 1. Source data of FRET assays can be found in Supplementary Data 2. Used plasmids and mass spectrometry raw files of binding site identification studies of purified MBP-SpsB are available upon request from the authors.

## Code availability
Custom code for performing simulation and analysis is available from the cited references in the paper. In-house analysis code (python code) is also available at *Zenodo*[69].

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

## Acknowledgements

We thank M. Wefelmeyer for helping with cloning. We thank M. Hitzenberger for building the initial model of SpsB and R. Zschau for technical support. Computer resources for this project have been provided by the NHR@FAU supercomputer facility at Regionales Rechenzentrum Erlangen (RRZE), Germany. We thank K. Bäuml and M. Wolff for their technical assistance. We thank D. Mostert for critical proof-reading of the manuscript. M.K.F. acknowledges a Kekulé stipend from "Fond der Chemischen Industrie" (FCI). Project was funded by Bundesministerium für Bildung und Forschung (BMBF) and VDI/VDE (Funding number: 16GW0265K). Figure 2e was created with Biorender.com.

## Author contributions

S-Y. Chen, M. K. Fiedler, T. Gronauer, S. Schneider, M. Zacharias and S. A. Sieber designed the project. S-Y. Chen performed all computational studies. M. K. Fiedler performed all wet lab experiments with the help of T. Gronauer, O. Omelko, S. Schneider and T. Wang. MBP-tagged SpsB used for binding site identification studies was purified by M-K. v. Wrisberg. M. K. Fiedler, S-Y. Chen, S. Schneider, M. Zacharias and S. A. Sieber wrote the manuscript for publication with input from all co-authors listed above.

## Funding

## Competing interests

The authors declare the following competing interests: S.A. Sieber is co-founder of smartbax limited. All other authors declare no competing interests.
