## [Transparent Peer Review file · Communications Biology]

Unraveling the mechanism of small molecule induced activation of *Staphylococcus aureus* signal peptidase IB

Corresponding Author: Professor Martin Zacharias

Figures originally included in the author's rebuttal have been redacted from this file.

Version 0:

Reviewer comments:

Reviewer #1

(Remarks to the Author)

In this work, the authors have done detailed study to try to discern the mechanism of action for activating *Staphylococcus aureus* signal peptidase IB (SpsB) with small molecules such as PK150, a known antibiotic. Their report includes mass spectrometry to probe which parts of the protein interact with small molecules such as PK150, preparation of mutants of SpsB which helped to delineate the residues that are important for the binding pocket and molecular dynamics simulations of the structure of SpsB without ligand and in the presence of ligand. They present evidence that the PK150 binds to a pocket adjacent to the active site and serves as an allosteric activator. Overall, the work is worth publishing. However, it is important first for the following concerns to be addressed.

Major Concerns

A significant concern is whether modifying a small molecule like PK150 by adding on a functional group to make it into an affinity-based probe will also modify the binding mode of the probe molecule relative to the parent molecule. The authors address this concern by...

II. 118-119. It is not clear that Figure 1c shows that "the collective motion of the globular domain is more coordinated when the substrate is associated". Best to add more detailed analysis of that point or delete it if there is no evidence.

In Figure 1, we can see that the lipid bilayer is not flat (rough but with a downward trend from left to right). The authors should check their work and provide an explanation about that (or redo after making sure that the system is set up and run properly).

I.136. The units for volume are Å³.

I.346. Here is a key error: "Undermining" should be "underlining".

II. 406-407. Meaning unclear. How can the substrate and the protein share the same structure?

I.419. Even when using SHAKE, most MD simulations use a timestep of 2 fs, not 4 fs. The authors should provide citations and explanation for how they can justify using such a large timestep without causing flaws in the simulations they run. E.g. Eur Phys J Spec Top. 2011 Nov 1; 200(1): 211–223 suggests that a large time step cannot be used.

Figures S6-S11 are problematic.

- The differences between the individual MD runs are very large.
- Individual runs also swing up and down very dramatically.
- "Time series" is not the correct technical term. Here and also on I.433. Do you mean "time dependence"? Or "Volume as a function of time"?

Minor Concerns

Abstract "poorly understood" should be "previously poorly understood".

Abstract "mutations". After a mutation has been carried out, the resultant product is a "mutant", not a "mutation".

I.64. It is unclear to what the term "these in depth analyses" references on this line. Rewrite for clarity.
I. 554. Units for the number?

Reviewer #2

(Remarks to the Author)

The authors describe a non-obligate heterotropic activator called PK150 ((4-Chloro-3-(trifluoromethyl)phenyl)-3-(2,2-difluorobenzo[d][1,3]dioxol-5-yl)urea).

Although the authors supply good chemical evidence for the PK150 binding site the manuscript is filled with speculation on the mechanistic ramifications of PK150 binding. The minor increase in activity (1.7 fold) is based on an assay performed with the enzyme in membranes and not backed up with purified and quantified enzyme. There is no attempt to measure enhancement of substrate affinity upon addition of PK150.

Below is a list of other problems with the analysis and manuscript.

Line 107: The results section of the manuscript leads with a MD simulation analysis with and without substrate. No description of the substrate is provided, one has to presume it is the synthesis peptide described in the methods section.

Line 134: A "channel" is mentioned, yet a detailed description of the structural details of this so-called channel is not provided.

Line 135: The descriptions of the pocket is extremely vague. Volume is mentioned yet a length is reported. Moreover these types of measurements depend greatly on the methods of analysis and the details for these measurements are lacking.

Line 155: It not accurate to refer to the S36 OG to K77 NZ distance as a hydrogen bond if it varies in length beyond reasonable hydrogen bond distance.

Line 191: PK150-P is significant more bulky than the original PK150. A discussion of how this could affect the binding is not addressed sufficiently.

The activity is measured only by normalized % cleavage (no thorough kinetic analysis is performed) and extremely small differences are observed. The measurement of activity is performed only with membranes with overexposed enzyme, not purifies and quantified enzymes.

P values are listed at the bottom of Figure legend 3. More detail on how these values were calculated is needed.

Line 276 (Figure 4 a, b, c): The labels on the plots and graphs are too small. The plots are not necessary in that the differences observed are not significant, the standard deviations look to be larger than the difference between bars.

Line 324: The role of the so-called "water channel" is speculative at best.

Line 325: The role of F67, Y75 and F158 motions having an effect on water access to the active site is also speculative.

Reviewer #3

(Remarks to the Author)

The authors report an experimental and MD simulation based analysis of the molecular mechanism underlying a small molecule-mediated activation of SpsB, a bacterial membrane-bound endoprotease with high medicinal relevance. The study is thereby based on a previous publication in which the authors found that PK150 activates SpsB, the exact molecular basis for this effect however remained elusive.

In the present study, the authors therefore used i) MD modeling to show that the hydration level of the active site plays a critical role for the enzymatic activity level of SpsB, ii) chemical proteomics to identify an allosteric binding site of PK150 nearby the active site residues and iii) a combination of MD analysis and biochemical assays to show that binding of PK150 to the allosteric site changes active site hydration, thereby triggering enzyme activation by a non-conventional mechanism.

Overall, these are important and relevant findings that in principle warrant publication in Commun Biol. As the study is also technically solid and well-written, I recommend its publication after some minor changes. I however would like to make the editor and the authors aware that I am no expert in the field of MD simulations and I therefore cannot comment informed on this part of the manuscript.

Minor points:

1) Fig. 2b – the authors report the Pearson coefficients in this analysis but do not really comment on them. In some cases, the Pearson coefficients are very low (e.g. 0.15 or even 0.05) which is usually interpreted as no correlation. Could the authors add an interpretation on these values to the MS? What do they mean for the assumption that the distances of F67, Y75 and F158 define the size of the allosteric pocket?

2) In general, many subfigures are much too small and therefore very difficult to read. For example, in Fig. 1c, I needed to use a 200% zoom to see any silver white arrows at all.

3) Fig. 2e - is it right that the upper subfigure (geometry representation) points to an area of the catalytic Hbond distance-water within 5Å of K77 diagram that is basically empty? If yes – could the authors add what should be learnt from this geometry presentation and its low level of adoption?

4) I could not find the excel file “S1_isoDTB_SpsB_PK150” attached to the SI (line 562).

5) The biochemical analyses of the mutants are important as they experimentally validate the MD-deduced activation mode. The authors have measured all biochemical effects in relation to PK150 via normalization to DMSO treatment. This makes sense. It could however also be interesting to compare the enzyme activity levels of the different mutants alone, i.e. without DMSO normalization and PK150 addition. Which mutant is more active/less active and how does this match to the proposed relevance of the “water channels” for the catalytic activity?

Author Rebuttal letter:

Garching, 13.5.2024

Dear Dr Huijuan Guo

Thank you for returning our manuscript entitled “Unraveling the mechanism of small molecule induced activation of Staphylococcus aureus signal peptidase IB” and the comments of the reviewers. In the following we like to comment on the concerns in a point-by-point response and indicate the changes and additions we have made to the manuscript. We include a version of the manuscript with all changes marked red.

Reviewers' comments:

Reviewer #1 (Remarks to the Author):

In this work, the authors have done detailed study to try to discern the mechanism of action for activating Staphylococcus aureus signal peptidase IB (SpsB) with small molecules such as PK150, a known antibiotic. Their report includes mass spectrometry to probe which parts of the protein interact with small molecules such as PK150, preparation of mutants of SpsB which helped to delineate the residues that are important for the binding pocket and molecular dynamics simulations of the structure of SpsB without ligand and in the presence of ligand. They present evidence that the PK150 binds to a pocket adjacent to the active site and serves as an allosteric activator. Overall, the work is worth publishing. However, it is important first for the following concerns to be addressed.

Major Concerns

1. A significant concern is whether modifying a small molecule like PK150 by adding on a functional group to make it into an affinity-based probe will also modify the binding mode of the probe molecule relative to the parent molecule. The authors address this concern by:

Author response: Thanks for the question, we agree that in general modifying the parent compound with additional functional groups e.g. photo crosslinker moiety may lead to deviations of binding modes. However, based on our previous publication by Le et al (Le P, Kunold E, Maccsics R, et al. Repurposing human kinase inhibitors to create an antibiotic active against drug-resistant Staphylococcus aureus, persists and biofilms. Nat Chem. 2020;12(2):145-158. doi:10.1038/s41557-019-0378-7) thorough structure activity relationship (SAR) studies have been performed that show that modifying the residues on ring 2 (see manuscript, Figure 2a) are tolerated, retain biological activity and still overactivate SpsB. The authors identified a strong correlation between antibiotic activity and SpsB target engagement/overactivation (see examples in the Figure shown below). Here, PK150-P shows a MIC value of 1.0 μ M in S. aureus NCTC 8325 compared to the frontrunner PK150 with a MIC of 0.3 μ M. The tested photoprobe PK150-P used in this study was originally used for standard affinity-based protein profiling (AfBPP) for mode of action and target identification studies and was successfully used to enrich SpsB as one of the main targets (next to the second validated target enzyme menG). Although the photoprobe PK150-P itself cannot be used in the FRET assay due to interferences, its derivative lacking the diazirine moiety (named 3-005) was found to activate SpsB comparably to PK150.

We thus conclude that PK150 and other derivatives like PK150-P do share identical binding sites with very similar mode of actions and binding modes to overactive SpsB. For thorough SAR studies, please check our

previous publication. (Le P, Kunold E, Macsics R, et al. Repurposing human kinase inhibitors to create an antibiotic active against drug-resistant *Staphylococcus aureus*, persisters and biofilms. *Nat Chem.* 2020;12(2):145-158. doi:10.1038/s41557-019-0378-7).

We added a sentence to the binding site identification part of the manuscript that the probe was previously used for target identification to make this more understandable for the reader (page 5).

Additionally, our observed binding site by isoDTB proteomic workflow is fully compatible with the docking and MD simulations, where ring 2 of PK150 is exposed to the solvent environment, thus sterically allowing modifications at this position.

We added a sentence to the manuscript that emphasizes the compatibility of experimental observations with our docking and MD simulations (page 6).

3. 118-119. It is not clear that Figure 1c shows that the collective motion of the globular domain is more coordinated when the substrate is associated. Best to add more detailed analysis of that point or delete it if there is no evidence.

Author response: Thanks for the comment. The observation of the collective motion of the globular domain was indeed based on visual inspection and a qualitative analysis. We agree with the reviewer that our statement is not sufficiently supported and the sentence has been removed in the revised version.

4. In Figure 1, we can see that the lipid bilayer is not flat (rough but with a downward trend from left to right). The authors should check their work and provide an explanation about that (or redo after making sure that the system is set up and run properly).

Author response: Thanks for the observation. The simulations were performed correctly. However, the impression of a downward trend was caused by the rendering procedure (slight tilting with respect to the x/y plane) in the VMD visualization software. We corrected the procedure and corrected the figure 1.

5. I.136. The units for volume are Å³.

Author response: Corrected.

I.346. Here is a key error: Undermining should be underlining.

Author response: Corrected.

6. 406-407. Meaning unclear. How can the substrate and the protein share the same structure?

Author response: Thanks for the question. An initial substrate-free structure is modelled by removing the substrate from the substrate-bound form and therefore the reference conformations of the protein part are identical. In the revised version the sentence is rewritten to reduce confusion.

7. I.419. Even when using SHAKE, most MD simulations use a timestep of 2 fs, not 4 fs. The authors should provide citations and explanation for how they can justify using such a large timestep without causing flaws in the simulations they run. E.g. *Eur Phys J Spec Top.* 2011 Nov 1; 200(1): 211-223 suggests that a large time step cannot be used.

Author response: Thanks for the important comment. On top of SHAKE constraint, we also used the hydrogen mass repartitioning scheme (HMR) to increase the time step by two folds. The HMR approach with a 4fs time step has been introduced by Roitberg and coworkers (*Journal of chemical theory and computation*, 11(4), 1864-1874. <https://doi.org/10.1021/ct5010406>, cited in our manuscript) and shown to reproduce results with the non-HMR with 2fs time step in both protein (*Journal of chemical theory and computation*, 11(4), 1864-1874. <https://doi.org/10.1021/ct5010406>) and membrane systems (*J. Chem. Inf. Model.* 2021;61(2):831-839. Doi: 10.1021/acs.jcim.0c01360) systems. It was also systematically analyzed by Piana and coworkers (Piana et al., *PNAS* (2013) 110, 5915-5920, <https://doi.org/10.1073/pnas.1218321110>) to optimize hydrogen repartitioning for the Anton special purpose MD supercomputer (D.E. Shaw laboratories). These authors use even larger time steps of 4.5 fs and 5 fs.

8. Figures S6-S11 are problematic.

Author response: Thanks for your comment on the supporting figures. We addressed your concerns individually below.

â€ The differences between the individual MD runs are very large.

Author response: For the enzyme in the holo and holo+PK150 simulations the RMSD is for most cases in the regime of 2-3 Å. However, a larger RMSD is seen in case of the apo and apo+PK150 simulations (now Figure S9). This is expected since the starting structure is not in the native form but a generated model structure by removing the substrate from the holo structures (because an experimental structure of the apo form is not available). For previous Figure S8 (now Figure S10) indeed quite large RMSD of substrate and ligand were measured because the simulations are aligned/superimposed with respect to the enzyme (protein) and therefore not only conformational changes of the substrate or ligand but overall movement with respect to the protein can result in large RMSD. Secondly, the RMSD we showed in Figure S10 was also large because we included

the mobile termini of the substrate (that not bound to the protein). In a revised Figure S10 we considered only the substrate part bound to the enzyme which is relevant for our study resulting now in much smaller RMSD vs. time.

â€ Individual runs also swing up and down very dramatically.

Author response: With the revised version of Figure S10 this is essentially only seen in Figure S11 (previous S9). As described above, the RMSD of substrate and ligand are measured after superposition of the protein onto the start structure. Hence, a large RMSD in the case of the PK150 indicates a significant overall movement of the ligand relative to the protein including partial or (in few cases) full dissociation. As discussed in the manuscript we see significant relative mobility of the PK150 ligand in the proposed binding groove especially in the apo-case (with no substrate bound to the enzyme).

â€ Time seriesâ is not the correct technical term. Here and also on l.433. Do you mean time dependence? Or Volume as a function of time?

Author response: We have changed the term to time evolution.

Minor Concerns

9. Abstract âpoorly understoodâ should be âpreviously poorly understoodâ.

Author response: We rephrased this part in the abstract: âOverall, our study elucidates a previously little investigated mechanism of enzyme activation and serves as a starting point for the development of future enzyme activators.â

10. Abstract âmutationsâ. After a mutation has been carried out, the resultant product is a âmutantâ, not a âmutationâ.

Author response: We corrected the phrasing to âmutantâ in the abstract.

11. l.64. It is unclear to what the term âthese in depth analysesâ references on this line. Rewrite for clarity.

Author response: We rephrased the part: âThus, most activators are discovered by coincidence or activity screens and often lack a firm mechanistic understanding of their mode of activation. However, in-depth mechanistic analyses are pivotal to advance the rational design of activators and catalyze their use in drug developmentâ

12. l. 554. Units for the number?

Author response: Thanks for the question, we checked this part and could not identify a missing unit.

Reviewer #2 (Remarks to the Author):

The authors describe a non-obligate heterotropic activator called PK150 ((4-Chloro-3-(trifluoromethyl)phenyl)-3-(2,2-difluorobenzo[d][1,3]dioxol-5-yl)urea).

1. Although the authors supply good chemical evidence for the PK150 binding site the manuscript is filled with speculation on the mechanistic ramifications of PK150 binding. The minor increase in activity (1.7 fold) is based on an assay performed with the enzyme in membranes and not backed up with purified and quantified enzyme. There is no attempt to measure enhancement of substrate affinity upon addition of PK150.

Author response: Thanks for the important comment. Contrary to inhibition, enzyme activation can show phenotypic effects already with increases of activity by only 10 to 20 % (Zorn, J. A. & Wells, J. A. Turning enzymes ON with small molecules. *Nature chemical biology* 6, 179-188 (2010), Bishop, A. C. & Chen, V. L. Brought to life: targeted activation of enzyme function with small molecules. *Journal of chemical biology* 2, 1-9 (2009).

In general, this manuscript represents a follow-up project on our previously published paper on PK150's mode of action in *Nature Chemistry* by Le et al (Le P, Kunold E, Maccsics R, et al. Repurposing human kinase inhibitors to create an antibiotic active against drug-resistant *Staphylococcus aureus*, persists and biofilms. *Nat Chem.* 2020;12(2):145-158. Doi:10.1038/s41557-019-0378-7) that thoroughly validated a phenotypic effect of SpsB overactivation by PK150 on *S. aureus*. They identified a strong correlation between antibiotic activity and SpsB overactivation leading to a validated uncontrolled secretion of proteins including members of the autolysin family. Most importantly, dysregulation of autolysin abundance is known to induce uncontrolled cell-wall degradation which was confirmed by mass spectrometry (MS), electron microscopy (EM), and autolysis experiments. Therefore, allosteric stimulation of SpsB activity e.g. by 170% fold at 10 μ M in *E. coli* induced SpsB membranes is sufficient to dysregulate *S. aureus* and thus represents a promising strategy for drug development against MRSA. We additionally want to emphasize that overactivation increases up to 250% at higher compound concentrations (see now manuscript Figure 3a).

To further address your concerns, the previous publication by Le et al thoroughly validated the assay in SpsB induced *E. coli* membranes to circumvent the following problems of working with purified full length SpsB. Since SpsB is a membrane-bound endopeptidase, it relies on its transmembrane anchor to be fully functional. The purification of full-length SpsB is extremely tedious and suffers from overall stability due to aggregation

and/or self-digestion, which is thus not suitable for larger validation studies. The use of truncated SpsB constructs lacking parts of their transmembrane segment is in general possible but shows a strong decrease in enzyme activity that can only be restored upon addition of non-ionic detergents/adjuvants mimicking the membrane environment (Rao C.V., S. et al. Enzymatic investigation of the *Staphylococcus aureus* type I signal peptidase SpsB â implications for the search for novel antibiotics. FEBS J. 276, 3222â3234 (2009)). This overall makes the system more artificial, and more assay interferences are likely to occur. Thus, the best way to overcome the mentioned problems was using a more stable system, that resembles the membrane environment and reliably measures enzyme activity and potential overactivation. The validation experiments of the FRET assay in *E. coli* induced membranes in the previous publication by Le et al. included purified and quantified full-length SpsB and endogenous *S. aureus* membranes resulting in overactivated enzyme as equally observed for SpsB induced *E. coli* membrane fractions. For more detailed information, see original publication by Le et al.

We added detailed information to the introduction of the manuscript to make the phenotypic effect on *S. aureus* by overactivation of SpsB clearer for the readers (page 3). We additionally added that the assay was previously validated in the results section.

Below is a list of other problems with the analysis and manuscript.

2. Line 107: The results section of the manuscript leads with a MD simulation analysis with and without substrate. No description of the substrate is provided, one has to presume it is the synthesis peptide described in the methods section.

Author response: Thank you for the comment. The sequence of the substrate is now provided in the results section for clarity (page 4).

3. Line 134: A "channel" is mentioned, yet a detailed description of the structural details of this so-called channel is not provided.

Author response: The term channel is removed to reduce confusion.

4. Line 135: The description of the pocket is extremely vague. Volume is mentioned yet a length is reported. Moreover, these types of measurements depend greatly on the methods of analysis and the details for these measurements are lacking.

Author response: In the updated manuscript, the description about the pockets is elucidated in more detail. Also, we mention the method with the MDpocket reference in both results and methods sections (page 4, 13).

5. Line 155: It not accurate to refer to the S36 OG to K77 NZ distance as a hydrogen bond if it varies in length beyond reasonable hydrogen bond distance.

Author response: In the updated manuscript, this distance is termed S36-K77 distance and illustrated in Figure 1d.

6. Line 191: PK150-P is significantly more bulky than the original PK150. A discussion of how this could affect the binding is not addressed sufficiently.

Author response: (see also response to a very similar comment 1.) of reviewer 1) Thanks for the question, we agree that in general modifying the parent compound with additional functional groups e.g. photo crosslinker moiety may lead to deviations of binding modes. However, based on our previous publication by Le et al (Le P, Kunold E, Macsics R, et al. Repurposing human kinase inhibitors to create an antibiotic active against drug-resistant *Staphylococcus aureus*, persists and biofilms. Nat Chem. 2020;12(2):145-158. doi:10.1038/s41557-019-0378-7) thorough SAR studies have been performed that show that modifying the residues on ring 2 (see Figure 2a) are tolerated, retain biological and overactivate SpsB. They identified a strong correlation between antibiotic activity and SpsB target engagement/overactivation (see Figure 1). Here, PK150-P shows an MIC value of 1.0 μM in *S. aureus* NCTC 8325 compared to the frontrunner PK150 with an MIC of 0.3 μM . The tested photoprobe PK150-P used in this study was originally used for standard affinity-based protein profiling (AfBPP) for mode of action and target identification studies and was successfully used to enrich SpsB as one of the main targets (next to the second validated target enzyme menG). Although the photoprobe PK150-P itself cannot be used in the FRET assay due to interferences, its derivative lacking the diazine moiety (named 3-005) was found to activate SpsB comparably to PK150 (212% (3-005) to 250% (PK150) overactivation compared to DMSO at 50 μM compound concentration) indicating similar binding modes of all these derivatives leading to overactivation of SpsB (see Figure below). We thus conclude that PK150 and other derivatives like PK150-P do share similar mode of actions and binding modes to overactive SpsB.

We added a sentence to the binding site identification part of the manuscript that the probe was previously used for target identification to make this more understandable for the reader (see response to reviewer 1).

Additionally, our observed binding site by isoDTB proteomic workflow is fully compatible with the docking and MD simulations, where ring 2 of PK150 is exposed to the solvent environment, thus allowing modifications at this position.

We added a sentence to the manuscript that emphasizes the compatibility of experimental observations with

our docking and MD simulations (see response to reviewer 1).

7. The activity is measured only by normalized % cleavage (no thorough kinetic analysis is performed) and extremely small differences are observed. The measurement of activity is performed only with membranes with overexposed enzyme, not purified and quantified enzymes.

Author response: Thanks for the question: Regarding the differences observed in this assay, contrary to inhibition, enzyme activation shows phenotypic effects already with increases of activity by only 10 to 20 % as mentioned in the answers above (Zorn, J. A. & Wells, J. A. Turning enzymes ON with small molecules. *Nature chemical biology* 6, 179-188 (2010), Bishop, A. C. & Chen, V. L. Brought to life: targeted activation of enzyme function with small molecules. *Journal of chemical biology* 2, 1-9 (2009). Our previously published paper on PK150's mode of action in *Nature Chemistry* by Le et al (Le P, Kunold E, Macsics R, et al. Repurposing human kinase inhibitors to create an antibiotic active against drug-resistant *Staphylococcus aureus*, persisters and biofilms. *Nat Chem.* 2020;12(2):145-158. doi:10.1038/s41557-019-0378-7) thoroughly validated a phenotypic effect of SpsB overactivation by PK150 on *S. aureus*. They identified a strong correlation between antibiotic activity and SpsB overactivation leading to a validated uncontrolled secretion of proteins including members of the autolysin family. Therefore, stimulation of SpsB activity e.g. by 1.7 fold at 10 μM in *E. coli* induced SpsB membranes can be viewed as a significant overactivation sufficient to dysregulate *S. aureus*. We additionally want to emphasize that overactivation increases up to 250% at higher compound concentrations (see now manuscript Figure 3a). Small differences observed for the mutants are necessary to conclude that the overall activation is not taking place anymore. We do agree that the inhibition effect observed for PK150 addition in the mutants is small to moderate (as we have already described the effect in the manuscript), however, we did not want to ignore our novel observation and performed MD simulations to identify an underlying reason for this behavior. Nonetheless, we want to emphasize that our main focus lies on elucidating the molecular mode of activation of PK150, which we were able to validate with our mutants.

Regarding the validation of the overall assay in *E. coli* induced membranes see answer to the following comments above and the original publication by Le et al. which included measurement of activity with purified enzyme as part of assay validation.

In conclusion, SpsB overexpressed in *E. coli* membranes shows an identical enzyme activation mechanism by PK150 compared to purified full length SpsB or endogenous *S. aureus* membranes. To account for potential background interferences of the FRET assay with the membrane, non-induced *E. coli* membranes that are not containing any SpsB were shown to have minor to no substrate cleavage activity and were included in each experiment as a control and subtracted in each experiment as basal activity before normalized activity was determined (previously done by Le et al and in current manuscript).

For further kinetic analyses of SpsB, we refer to the published papers by Rao et al and Le et al (Rao C.V., S. et al. Enzymatic investigation of the *Staphylococcus aureus* type I signal peptidase SpsB - implications for the search for novel antibiotics. *FEBS J.* 276, 3222-3234 (2009)

We do agree that quantification of SpsB in each individual mutant is not possible with our FRET assay performed in *E. coli* membranes, as solely the amount of used membrane was kept constant for each mutant and each experiment and that enzyme activity might be impaired in some mutants compared to the wildtype. However, the focus was to show that the overall overactivation effect is not observed when crucial residues are mutated that were previously identified by MD simulations and hence, validate the performed MD simulations. To circumvent a misleading data interpretation, we normalized the data to each DMSO control separately for each mutant and do not compare the mutants among each other. Showing unnormalized data would result in a misleading interpretation of the data as one might directly compare the substrate cleavage rates among the mutants concluding that this directly correlates with the enzyme activity of each mutant which would be not correct since we are not able to quantify the amount of SpsB in the membranes.

We added this information on how and why we present the data as shown to the methods section and figure legend (3f) to make it clearer for the reader. We additionally added detailed information to the introduction and results section of the manuscript to make the phenotypic effect on *S. aureus* by overactivation of SpsB clearer for the readers. We additionally added to the results section that the assay with *E. coli* membranes was thoroughly validated in the publication by Le et al.

8. P values are listed at the bottom of Figure legend 3. More detail on how these values were calculated is needed.

Author response: Thanks for the comment, this has been corrected and made clear. P values were calculated by GraphPad Prism 10 by a two-tailed Student's t-test of compound versus DMSO treated samples, performed individually for each mutant as we do not compare the mutants among each other (page 19).

9. Line 276 (Figure 4 a, b, c): The labels on the plots and graphs are too small. The plots are not necessary in that the differences observed are not significant, the standard deviations look to be larger than the difference between bars.

Author response: Labels are now larger due to rearrangement of the figure. To 4a: This dose-dependent plot

shows very accurate sigmoidal behaviour across several biologically independent experiments, suitable for IC50 or EC50 measurements of compounds and are reaching either as expected either 0% activity or an overactivation plateau (R^2 (wt A4) = 0.95, R^2 (F158 A4) = 0.92, R^2 (wt PK150) = 0.86, R^2 (F158 PK150) = 0.91) To 4b, the violin plots showing all data points are replaced by the average values \pm s.d of the mean ($n=5$) to demonstrate the deviation across independent simulations. To 4c, the data is limited by the computational resources and we can only demonstrate the dissociation incidences within five 1-microsecond long simulations. Ideally better statistics can be obtained by more replicates and longer simulations. However, given that a tightly bound ligand is not expected to dissociate from its binding pocket, as we have shown in the holo+PK150 systems, multiple early dissociation events in the apo+PK150 systems suggest that the binding of PK150 is weakened without the presence of the substrate.

10. Line 324: The role of the so-called "water channel" is speculative at best.

Author response: The term water channel has been removed in the updated manuscript.

11. Line 325: The role of F67, Y75 and F158 motions having an effect on water access to the active site is also speculative.

Author response: In the simulations of the wt SpsB enzyme, we observed that the water access mildly correlates with the distance between F67, Y75, and F158 and thus discussed potential functional roles of these amino acids. Indeed, in the mutant simulations we also found out that mutating these residues to alanine can increase the active site water accessibility to different degrees. This is now depicted in Figure S7, S8 in the updated manuscript (see also page 8).

Reviewer #3 (Remarks to the Author):

The authors report an experimental and MD simulation based analysis of the molecular mechanism underlying a small molecule-mediated activation of SpsB, a bacterial membrane-bound endoprotease with high medicinal relevance. The study is thereby based on a previous publication in which the authors found that PK150 activates SpsB, the exact molecular basis for this effect however remained elusive.

In the present study, the authors therefore used i) MD modeling to show that the hydration level of the active site plays a critical role for the enzymatic activity level of SpsB, ii) chemical proteomics to identify an allosteric binding site of PK150 nearby the active site residues and iii) a combination of MD analysis and biochemical assays to show that binding of PK150 to the allosteric site changes active site hydration, thereby triggering enzyme activation by a non-conventional mechanism.

Overall, these are important and relevant findings that in principle warrant publication in *Commun Biol*. As the study is also technically solid and well-written, I recommend its publication after some minor changes. I however would like to make the editor and the authors aware that I am no expert in the field of MD simulations and I therefore cannot comment informed on this part of the manuscript.

Minor points:

1) Fig. 2b - the authors report the Pearson coefficients in this analysis but do not really comment on them. In some cases, the Pearson coefficients are very low (e.g. 0.15 or even 0.05) which is usually interpreted as no correlation. Could the authors add an interpretation on these values to the MS? What do they mean for the assumption that the distances of F67, Y75 and F158 define the size of the allosteric pocket?

Author response: In the updated version, we moved the correlation plot to the supporting information and only mentioned a "moderate correlation" between the allosteric pocket size and the F158-F67 distance and F158-Y78 distance. The claim about the amino acids defining the pocket size is also removed.

2) In general, many subfigures are much too small and therefore very difficult to read. For example, in Fig. 1c, I needed to use a 200% zoom to see any silver white arrows at all.

Author response: Thanks for the comment, as part of the overall revisions, we merged and rearranged the previous figure 1 and figure 2 to one new figure (figure 1 in the revised manuscript). Additionally, we rearranged the previous figure 4 to enhance readability (can now be found as figure 3 and 4 in the revised manuscript).

3) Fig. 2e - is it right that the upper subfigure (geometry representation) points to an area of the catalytic Hbond distance-water within 5Å of K77 diagram that is basically empty? If yes - could the authors add what should be learnt from this geometry presentation and its low level of adoption?

Author response: Thanks for the question. The population of such geometry (5ns-averaged water count \approx 1.2 and S36-K77 distance \approx 2.5 Å) in the holo simulations is 8.5% (now included also in the manuscript figure). Although the population is low, our data suggested that the larger S36-K77 distance is conditioned with the

water accessibility ($8.5\% / (2.4\% + 8.5\%) = 79.0\%$). In addition, in the revised manuscript we demonstrated a high correlation between the active site water accessibility and S36-K77 distance across 20 systems (Pearson R: 0.88, Figure S8).

4) I could not find the excel file "S1_isoDTB_SpsB_PK150" attached to the SI (line 562).

Author response: Thanks for the important observation, we realized that this file got lost during the submission process. We uploaded the respective excel sheet with our resubmission files.

5) The biochemical analyses of the mutants are important as they experimentally validate the MD-deduced activation mode. The authors have measured all biochemical effects in relation to PK150 via normalization to DMSO treatment. This makes sense. It could however also be interesting to compare the enzyme activity levels of the different mutants alone, i.e. without DMSO normalization and PK150 addition. Which mutant is more active/less active and how does this match to the proposed relevance of the "water channels" for the catalytic activity?

Author response: Thanks for this comment, we do agree that this is in general interesting to investigate. Our previous publication by Le et al (Le P, Kunold E, Macsics R, et al. Repurposing human kinase inhibitors to create an antibiotic active against drug-resistant *Staphylococcus aureus*, persists and biofilms. *Nat Chem.* 2020;12(2):145-158. doi:10.1038/s41557-019-0378-7) thoroughly validated the assay in SpsB induced *E. coli* membranes to overcome certain problems of working with purified full length SpsB. Since SpsB is a membrane-bound endopeptidase, it relies on its transmembrane anchor to be fully functional. The purification of full-length SpsB is extremely tedious and suffers from overall stability due to aggregation and/or self-digestion, which is thus not suitable for larger validation studies (e.g. potential self-digestion during assay). The use of quantified, truncated SpsB constructs lacking parts of their transmembrane segment is in general possible but show a huge decrease in enzyme activity that can only be restored upon addition of non-ionic detergents/adjuvants mimicking the membrane environment (Rao C.V., S. et al. Enzymatic investigation of the *Staphylococcus aureus* type I signal peptidase SpsB - implications for the search for novel antibiotics. *FEBS J.* 276, 3222-3234 (2009)). This overall makes the system more artificial, and more assay interferences are likely to occur.

Thus, the best way to overcome the mentioned problems was using a more stable system, that resembles the membrane environment and reliably measures enzyme activity and potential overactivation. The validation experiments of the FRET assay in *E. coli* induced membranes in the previous publication by Le et al. included purified and quantified full-length SpsB and endogenous *S. aureus* membranes resulting in overactivated enzyme as equally observed for SpsB induced *E. coli* membrane fractions.

However, with our FRET assay performed in *E. coli* membranes, quantification of SpsB in each individual mutant is not possible, as solely the amount of used membrane was determined by BCA assay and kept constant for each mutant, not necessarily representing the exact amount of SpsB in each membrane fraction. The overall focus here was to show that the overactivation effect is not observed when crucial residues are mutated that were previously identified by MD simulations and hence, validate the performed simulations. To circumvent a misleading data interpretation, we normalized the data to each DMSO control separately for each mutant and do not compare the mutants among each other. Showing unnormalized data would result in a misleading interpretation of the data as one might directly compare the substrate cleavage rates among the mutants concluding that this directly correlates with the enzyme activity of each mutant which would be not correct since we are not able to quantify the amount of SpsB in the membranes. However, we do agree that enzyme activity might be impaired in some mutants compared to the wildtype and studying the individual mutants in terms of activity in comparison to each other will be focus of future work.

To make it clearer for the readers, we added this information on how and why we present the data as shown to the SI part and the figure legend (3f) of the FRET assay.

We hope that with the additions and changes we have made to the manuscript it is now acceptable for publication in the *Communications Biology*.

Yours sincerely,
Martin Zacharias

Version 1:

Reviewer comments:

Reviewer #1

(Remarks to the Author)

The work has been significantly revised and the authors have taken care to address all the concerns of the reviewers. The work is now nearly acceptable for publication. Below are some small errors to fix.

Some of the key new sentences are hard to interpret because of missing commas. Here is an attempt, but the authors should confirm their meaning.

E.g., "However, in MD simulations, for F67A, Y75A, and 3A mutants allosteric pocket expansion was not observed in holo SpsB blocking the activation of SpsB (Figure 3b and Figure 4, top). For F158A, the increased size of the allosteric pocket enhances the likelihood of PK150 binding to the apo-form with a deviated binding mode perturbing substrate binding."

Should be:

"However, in MD simulations, for F67A, Y75A, and 3A mutants, allosteric pocket expansion was not observed in holo SpsB, blocking the activation of SpsB (Figure 3b and Figure 4, top). For F158A, the increased size of the allosteric pocket enhances the likelihood of PK150 binding to the apo-form with a deviated binding mode, perturbing substrate binding."

And

"Showing unnormalized data would result in a misleading interpretation of the data as one might directly compare the substrate cleavage rates among the mutants concluding that this directly correlates with the enzyme activity of each mutant which would be incorrect since quantification of SpsB (wildtype or mutant) in each membrane cannot be performed."

Should be:

"Showing unnormalized data would result in a misleading interpretation of the data, as one might directly compare the substrate cleavage rates among the mutants, concluding that this directly correlates with the enzyme activity of each mutant, which would be incorrect since quantification of SpsB (wildtype or mutant) in each membrane cannot be performed." Or they may wish to otherwise improve the sentence to make their meaning clear.

Since Figs S2 and S3 are different but have identical captions, please check/correct the caption for Fig S2. (The caption states "Figure S2. Water population in the allosteric pocket (circled red) and in the substrate pocket (circled orange)." But no red and orange circles are visible in Figure S2 itself.)

Main text, p.7., and Figure S6 caption. Please confirm that the P values are all "greater than" except the last one in the statement "P values: $P > 0.1234$ (ns), 0.0332 (*), 0.0021 (**), 0.002 (***), < 0.0001 (****)". Normally, for stat significance, it would be best to have the P values less than a certain number such as 0.002, not " > 0.002 ".

Figure S7 caption. Error: "and K77 S36-K77 distance" should be "and S36-K77 distance". "average amount of water molecules" should be "average number of water molecules".

Reviewer #3

(Remarks to the Author)

I am fine with all made corrections/additions that addressed adequately my concerns. The paper can now be published in its present form.

Author Rebuttal letter:

Garching, 21.6.2024

Reviewers' comments:

Reviewer #1 (Remarks to the Author): The work has been significantly revised and the authors have taken care to address all the concerns of the reviewers. The work is now nearly acceptable for publication. Below are some small errors to fix.

1. Some of the key new sentences are hard to interpret because of missing commas. Here is an attempt, but the authors should confirm their meaning.

E.g., "However, in MD simulations, for F67A, Y75A, and 3A mutants allosteric pocket expansion was not observed in holo SpsB blocking the activation of SpsB (Figure 3b and Figure 4, top). For F158A, the increased size of the allosteric pocket enhances the likelihood of PK150 binding to the apo-form with a deviated binding mode perturbing substrate binding."

Should be:

"However, in MD simulations, for F67A, Y75A, and 3A mutants, allosteric pocket expansion was not observed in holo SpsB, blocking the activation of SpsB (Figure 3b and Figure 4, top). For F158A, the increased size of the allosteric pocket enhances the likelihood of PK150 binding to the apo-form with a deviated binding mode, perturbing substrate binding."

Response: We corrected the sentence according to the suggestions of the reviewer.

2. And

"Showing unnormalized data would result in a misleading interpretation of the data as one might directly

compare the substrate cleavage rates among the mutants concluding that this directly correlates with the enzyme activity of each mutant which would be incorrect since quantification of SpsB (wildtype or mutant) in each membrane cannot be performed.

Should be:

Showing unnormalized data would result in a misleading interpretation of the data, as one might directly compare the substrate cleavage rates among the mutants, concluding that this directly correlates with the enzyme activity of each mutant, which would be incorrect since quantification of SpsB (wildtype or mutant) in each membrane cannot be performed. Or they may wish to otherwise improve the sentence to make their meaning clear.

Response: We corrected the sentence according to the suggestions of the reviewer.

3. Since Figs S2 and S3 are different but have identical captions, please check/correct the caption for Fig S2. (The caption states "Figure S2. Water population in the allosteric pocket (circled red) and in the substrate pocket (circled orange)." But no red and orange circles are visible in Figure S2 itself.)

Response: We thank the reviewer for carefully checking the MS and spotting this error. We corrected the captions of Figures S2 and S3 accordingly.

4. Main text, p.7., and Figure S6 caption. Please confirm that the P values are all "greater than" except the last one in the statement "P values: $P > 0.1234$ (ns), 0.0332 (*), 0.0021 (**), 0.002 (***), <0.0001 (****)". Normally, for stat significance, it would be best to have the P values less than a certain number such as 0.002, not " >0.002 ".

Response: We corrected the sentence on page 7 (and in The Figure S6 caption) to confirm the P values.

5. Figure S7 caption. Error: "and K77 S36-K77 distance" should be "and S36-K77 distance". "average amount of water molecules" should be "average number of water molecules".

Response: Figure S7 caption has been corrected.

We hope that with the additions and changes we have made to the manuscript it is now acceptable for publication in the Communications Biology.

Yours sincerely,
Martin Zacharias
